# Preservation of Dinoflagellate Cysts in Different Oxygen Regimes: Differences in Cyst Survival between Oxic and Anoxic Natural Environments

Agneta Persson [1,*,†] and Barry C. Smith [2]

1   Department of Marine Ecology, Göteborg University, P.O. Box 461, SE-405 30 Göteborg, Sweden
2   Northeast Fisheries Science Center, Milford Laboratory, National Oceanic and Atmospheric Administration, National Marine Fisheries Service, Milford, CT 06460, USA
*   Correspondence: agnetapersson77@gmail.com; Tel.: +46-70-650-6227
†   Current address: Smedjebacksvägen 13, SE-771 90 Ludvika, Sweden.

**Abstract:** This quantitative dinoflagellate cyst study reveals an enormous difference in survival rates in oxygenated versus anoxic sediments. Replicate samples of concentrated natural dinoflagellate cysts with the same initial species composition ($1.4 \times 10^4$ resting cysts·cm$^{-3}$ sediment, 61% filled with live-appearing contents) were placed in bags of 20 μm plankton screen. Replicate bags containing 10.0 cm$^{-3}$ concentrated cyst samples were placed on the seafloor in different environments in Long Island Sound, USA (anoxic and oxygenated), as well as refrigerated in test tubes in the laboratory. Three sets of 15 bags were placed in each environment. Once every year for four consecutive years, three bags were recovered from each set, and the contents were analyzed by cyst counting and germination experiments. An enormous difference in preservation potential between samples in oxygenated versus anoxic environments was revealed. The number of dinoflagellate cysts decreased abruptly within the first year in the oxygen-rich environment; living cysts became very rare (only 5% remained) and also empty walls of cysts disappeared (20% of total cysts remained). In anoxic sediment samples, living cysts also decreased significantly with time, but less quickly. After 1 year, 35% of the living cysts in the anoxic environment and 70% of the living cysts refrigerated in test tubes remained intact. After 4 years, 21% of the cysts with contents in the anoxic environment remained, and 31% in test tubes. The empty cyst walls remained intact for a longer time under anoxic conditions, especially of species known to fossilize well. Germination experiments showed that cysts with live-appearing contents were likely alive, because species with identifiable live-appearing cysts were also identified as vegetative cells in corresponding slurry cultures. The cyst assemblage was dominated by Protoperidinaceae, Dipolopsalidaceae, and Gonyaulacaceae. Of special interest is the ichthyotoxic *Margalefodinium polykrikoides*, the bloom-forming *Peridinium quinquecorne*, which has an undescribed resting cyst, and a previously undescribed *Krypoperidinium* species. The results show greater preservation of dinoflagellate cysts in "dead-zone sea bottoms" and may also provide an answer to the question of the absence of cyst beds in an area despite observed sedimentation of dense blooms.

**Keywords:** dinoflagellate; resting cyst; preservation; hypoxia; sediment

## 1. Introduction

Dinoflagellates are important primary producers and micro-zooplankton, constituting a major component of the base of food webs in freshwater and marine environments globally [1,2]. They phylum is evolutionarily very old and comprises organisms with vastly different sizes and life strategies. Even though the majority of species are harmless, these protists are well known for causing algal blooms and accumulating in amounts coloring the water red-brown ('red tides'). Blooms are harmful when the dinoflagellate produces toxins, such as those causing paralytic (PSP), diarrhetic (DSP), neurotoxic (NSP),

or azaspiracid (AZP) shellfish poisoning, and ciguatera fish poisoning (CFP) in humans [3]. They are also considered harmful when producing toxins affecting fish (ichtyotoxins), and when large cell numbers cause hypoxia during decomposition at bloom termination (e.g., [4]). Dinoflagellates are, with few exceptions, haploid and multiply through binary division. Life strategies are diverse, and this paper narrows the view to dinoflagellates in temperate coastal areas that produce dormant resting cysts sized 20–100 μm that sink to the sediment. When life cycles of resting cysts producing dinoflagellates in temperate areas have been studied in detail, it has given rise to the general view that the dormant resting stage is formed through sexual fusion of two gametes into a diploid zygote. The zygote transforms into a resting cyst and sinks to the seafloor to overwinter [5]. The resting cyst is the only fossilizable part of a dinoflagellate, and only when the wall is composed of resistant material [6–8]. In temperate areas, a large proportion of dinoflagellate species are known to produce resting cysts (28% in Sweden, [9], compared to 13–16% of all, [10]). The geological record of dinoflagellates is highly selective, preserving only resistant resting cysts, and, of these, many are morphologically indistinct. Only half have a known fossil record [10]. Many well-known toxic dinoflagellates have a recognizable resting cyst stage. Examples from the area are *Alexandrium* species (producing paralytic shellfish poisoning toxins, e.g., [11]) and *Margalefodinium polykrikoides*, which is ichthyotoxic, causing harm to fish and shellfish [12].

Dinoflagellate resting cysts possessing a resistant wall are well preserved in anoxic sea bottoms [13]. They have a mandatory resting period lasting a few weeks to a couple of months, depending upon species [5]. The cyst remains quiescent until the environment becomes favorable, and then it germinates. The zygote divides to become haploid, free-living vegetative cells that continue to grow by binary fission; if conditions are favorable, its season ends with a bloom. Dinoflagellates constitute important parts of marine and freshwater food webs. Half of the species are photosynthetic primary producers and the other half heterotrophic, equipped with different elaborate structures for catching prey [2,14–17] and sometimes with complex eye spots [18]. Many photosynthesizing dinoflagellates are mixotrophic, catching prey in addition to possessing chloroplasts [2]. Hetero- and mixotrophic dinoflagellates belong to the microfauna, consuming primary producers (sometimes larger than themselves) and being consumed by the meiofauna (rotifers, ciliates, copepods, etc.) in the trophic structure leading to large fish and mammals.

In anoxic environments, dinoflagellate resting cysts remain quiescent awaiting a germination opportunity [19]. They may remain alive for over 100 years [20]. We previously showed that different animals graze and digest dinoflagellate resting cysts [21–25]. In [22], toxic *Alexandrium fundyense* (now *A. catanella*) cysts were digested by oysters, and toxins accumulated in oyster soft tissues. The lack of grazers in anoxic environments likely leads to higher preservation of cysts; unable to germinate, they remain alive [26].

There is a great difference in microbial and chemical characteristics between oxygen-free and oxygen-rich environments. Aerobic microorganisms are more efficient decomposers of organic material because of higher respiration rates compared to anaerobic microorganisms [27]. Prokaryotes in anoxic environments are dominated by facultative anaerobic fermenting bacteria for the initial degradation [28]. Sulfate-reducing bacteria and methanogenic archaea perform the terminal steps of degradation, depending upon metabolites produced in fermentation processes [28,29]. When organic material decomposes in an anoxic environment, organic acids are produced in different steps [28,29]. This can affect cysts with an outer wall of calcite since calcium carbonate is dissolved at low pH; resting cysts with outer walls of calcite are common in oxygen-rich sediments (e.g., in sand), but more rarely found in organic-rich black mud [30]. *Scrippsiella* spp. cysts without calcite crystals, but otherwise intact, are often found in anoxic mud ([31,32], and personal observations from the Swedish west coast and Long Island Sound, USA). The thin inner sporopollenin wall allows cysts to survive for some time without crystals, but they may be wrongly identified (discussed in [33]). Calcareous cysts are most often not found in the

fossil record because of palynological preparation methods using acids (HCl, HF, acetic acid, and $H_2SO_4$ [19,34]).

There is a well-known difference in cyst abundance and species composition between oxygenated and anoxic sediments (e.g., [35–38]). Paleontologists routinely sample anoxic sediments because untouched layers of sediments allow dating of layers by radiometric methods. Biologists often make surveys, e.g., along coasts, and interpret differences in cyst occurrences as differences in production or accumulation, not as differences in preservation. Sometimes a "mixed layer" is described on top of anoxic layered sediment. Here, different animals consume and mix the sediment (containing cysts), before the cysts that survive the chewing, grinding, and the digestive fluids of one animal after the other as intact live cysts or as whole or broken empty cyst walls, are finally deposited in the anoxic, deeper layers.

Grazing reduces the numbers of both living cysts and empty cyst walls and changes the species composition of dinoflagellate cysts [21–23,25,32,39–41]. Different grazers affect the species composition differently [21] using different feeding modes and types of digestion. Grazing by filtering and deposit-feeding macrofauna is, thus, an important factor that shapes the dinoflagellate cyst seed bank; however, what differences do we find between anoxic and oxygenated sediments when excluding macrofauna? Differences in microbial degradation between the environments may be important, but we must also consider the abundant micro- and meiofauna in oxygenated sediments. These small heterotrophic organisms (including many dinoflagellates) have rapid generation times and are important contributors to nutrient cycling and provision of food to higher trophic levels [42,43]. Many have resting stages and germinate in slurry cultures of sediment [44,45] and may also feed on cysts. Kodrans-Nsiah et al. [46] and Gray et al. [47] showed that species-selective degradation of dinoflagellate resting cysts in oxygenated natural environments is a rapid process that transforms dinoflagellate cyst concentrations and assemblages. They used palynological methods and did not study living dinoflagellate cysts, but showed that early diagenesis takes place rapidly under aerobic conditions and, therefore, cannot be neglected on any timescale. A 24–57% reduction in cyst numbers was found in oxygenated conditions in the 15 month study by Kodrans-Nsiah et al. [46], and Gray et al. [47] found a 32% reduction in cyst numbers during the first year in oxygenated conditions in their 5 year study. No detectable reduction in cyst numbers was found in anoxic conditions in either study. Zonneveld et al. [48] reported selective anaerobic degradation of dinoflagellate cysts in a palynological study of a uniquely layered core, where the time frame was hundreds and thousands of years.

Many human activities favor the formation of harmful algal blooms. Nutrients from farming, industry, cities, and traffic contribute to increased nutrient loads in coastal waters and increased growth of phytoplankton [49,50], as well as species-specific shifts toward more toxic species [49]. Toxic phytoplankton can also increase in proportion relative to more palatable species because of terrestrial herbicide pollution [51]. Overwintering resting stages are well preserved in anoxic sediments, and this contributes to increased numbers as "dead-zone sea bottoms" continue to spread with eutrophication [52]. The effects of human activities on nature can lead to a cycle of harmful algal blooms, which, to a certain extent, is self-accelerating; nutrients are well preserved in anoxic seafloors, as are resting stages. When harmful species have a competitive advantage, they use more of the resources, increase in proportion, produce more resting stages that are preserved, etc. The recent concern about globally increasing temperatures is relevant to the ecology of dinoflagellate blooms, as increased surface-water temperatures intensify water stratification, which is favorable for dinoflagellate bloom success [53] and, consequently, cyst production. High cyst yields result in larger cyst seed banks and increased harmful bloom risk. Increasing storm frequencies would elevate more benthic resting stages into surface waters from anoxic seed banks, causing blooms.

The research reported here compares the preservation and survival of living dinoflagellate cysts in oxygenated versus anoxic environments, excluding the action of grazing by macrofauna.

## 2. Materials and Methods

*2.1. Sampling*

### 2.1.1. Sample Origin

The experiment was performed on the east coast of the USA, at NOAA/NMFS Milford Laboratory, Connecticut. A large sample (20 US gallons = 76 L) of black anoxic mud was collected on 23 October 2008 by divers from a dredge pit in Morris Cove, New Haven, CT (41°15.556 N, 72°53.926 W) at 6.7 m depth. The 200 × 750 m pit was created in the 1950s via removal of 765,000 m$^3$ material for use as road base. It is an irregular depression ca 7 m deeper than the surrounding cove and known to collect fine-grained material (silt/clay); the pit has poor biological conditions with methane production and sulfur oxidation. In 2000, ca. 14,000 m$^3$ of organic-rich surface sediment dredged from the adjacent US Coast Guard station was deposited in the pit. It also receives natural deposition of fine-grained material, estimated to be 0.5 to 0.8 cm per year. A thorough description of the Morris Cove "borrow" pit is given in [54].

### 2.1.2. Sample Acquisition and Concentration

The vessel for sampling was the "Seahawk", belonging to the State of Connecticut, Aquaculture Division. Four 5 gallon (19 L) buckets were brought to the bottom by divers and scooped into the mud bottom. The divers finished filling the buckets by manually scooping mud into the buckets. The material consisted of sediments from the surface to 10 cm below. Once full, the bucket tops were secured, and the buckets were returned to the boat. Buckets were kept in darkness at 4 °C until treated as follows: on 24 October, a cyst-concentrating apparatus developed previously (a 152.5 × 44.5 × 38 cm fiberglass tank with a slanted screen, described in [25]) was set up and run for 72 h at 6.5–8.0 °C in darkness (covered with black plastic). In this apparatus, cysts become concentrated with minimal disturbance, in darkness and at a low temperature, to avoid germination. The apparatus concentrates dinoflagellate cysts from large amounts of sediment (concentration factor 376× for living cysts in [25]). Before addition to the apparatus, the sediment sample was sieved through 1 mm mesh and mixed to a thin slurry with cold (~6 °C) filtered seawater. The intake tube to the pump was standing atop an inverted 200 μm sieve in the bucket to prevent clogging. Thus, the sediment was introduced into the apparatus through a manifold at the deep end, and the flow rate was 1.5 L·min$^{-1}$. Cysts floated slowly toward the outlet in the shallow end and settled on the slanted screen according to density and morphological features. Each day, more sediment was added until all four 19 L buckets were added on 26 October (except 1 L of the original sample that was saved). The cyst fraction was recovered from the slanted screen by pipetting on 27 October (from the area 0–50 cm from the outlet, after determining this as the location of maximal cyst content in the sorter by microscopy), centrifuged (Beckman, Model J-6B) for 15 min at 2012× *g* in twelve 1 L flasks and placed cold. The water was decanted, and the sediment from all flasks (2110 mL) was combined into one 4 L container. This concentrated cyst sample was stored in darkness at 4 °C.

### 2.1.3. Preparation and Deployment of Bags Containing Concentrated Cyst Samples

The preparation was performed in as dark and cold conditions as possible; no lights were turned on in the laboratory, and everything was kept on ice under black plastic as much as possible. The concentrated cyst fraction was mixed thoroughly and placed in bags (rectangular with 9 × 11 cm inner measurement) made from 20 μm Nitex nylon plankton screen. Seams were sealed with aquarium silicone after sewing to ensure the containment of all intact dinoflagellate cysts 20 μm and larger. Each bag was filled with 10.0 mL of sediment, sealed, and sewn to ropes, three on each rope (Figure 1). At the end of each rope, a small lead weight (approximately 3 g) was attached to ensure that bags could not float. Five sets of triplicate bags were attached to a rope attached to a pole, making in total six sets of 15 bags to be deployed at one anoxic and one oxygenated site (three in each environment). Furthermore, fifteen 15 mL polypropylene test tubes were filled with the

same amount (10.0 mL) of the well-mixed concentrated sediment sample and placed in a refrigerator at 4 °C. One extra set of three bags was placed in a jar within 1 L of the original unsorted sample remaining and placed in the same refrigerator until September 2014, when the experiment ended. The remaining sorted anoxic sediment was also stored in the same refrigerator. Until deployment, bags (marked with embossed labels and attached to ropes as described) were kept in plastic bags filled with filtered seawater in a refrigerator (4 °C). They were deployed on the seafloor attached to poles equipped with radio transmitters (Sconotronic coded transmitters CT-82-2-E), by divers, on 5 November 2008. The bags were placed in order as in Figure 1 and covered with ca. 10 cm of adjacent sediment. The anoxic site was the same as used for sampling of the original sample (Morris Cove borrow pit, New Haven, CT (41:15.556 N, 72:53.926 W), depth: 8.5 m. The oxygenated site was situated outside New Haven harbor at 41:13.307 N, 72:56.597 W, depth: 10.7 m. Bags were spaced ~20 cm apart at 10 cm below the sea floor surface.

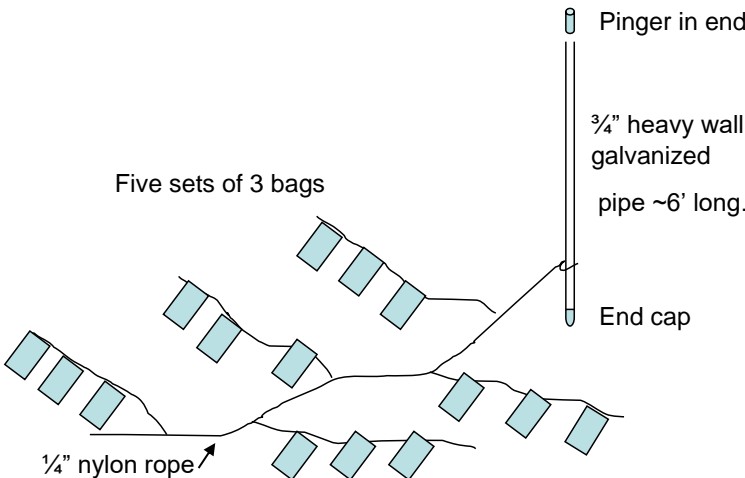

**Figure 1.** Experimental setup. Bags containing concentrated cysts were deployed at the sea bottom.

### 2.1.4. Preparation of Permanent Microscope Slides for Start Values

Permanent microscope mounts for counting of start values were made according to the method described in [9,55]. For start values, 10 mL samples were sonicated (Sonicor 5C-50; Sonicor instrument corporation, Copiaque, NY, USA) for 5 min, sieved with filtered seawater using a plant sprayer to retain the 20–100 μm fraction, sonicated for 2 min again, and sieved on the 20 μm sieve. The material was rinsed into a 200 mL volumetric flask, and volume was adjusted to 200.0 mL. Next, 10.0 mL was taken and sieved with distilled water on the 20 μm sieve, rinsed into a glass centrifuge tube (narrow bottom), and left standing in the refrigerator (4 °C, darkness) overnight. The sieved sediment samples constituted 0.18 mL in each tube when the water was removed. This sample was well mixed, and, from each tube, subsamples of 10 μL were taken to make permanent microscope mounts. Mounts were made; then, to each marked microscope slide on a 50 °C heater, a drop of glycerin was added, and, to this drop, the sample was added. The sample was mixed into the glycerin for an even distribution, and, when the water had evaporated, a cover glass was placed on top and the sample was sealed with beeswax. This method, which involves mounting of nontreated but rinsed cysts in glycerin and sealing with beeswax, preserves cysts with the morphology intact, i.e., contents remain visible and live-looking when cysts are alive at mounting. Thus, filled cysts with live-looking contents can be counted separately from empty cyst walls even after a long time of storage.

### 2.1.5. Dry Weight Measurement

Dry weight (DW) measurements: 10 mL samples (corresponding to the material in each bag and test tube) were dried to constant weight. The sediment concentrated by sorting, used in bags and test tubes in the experiment, had a dry weight of 38.7% ± 0.1%,

and the original unsorted sample had 37.2% ± 0.1% DW. The sediment within each bag and test tube corresponded to 3.9 ± 0.1 g DW concentrated sediment. Dinoflagellate cysts and other particles larger than 20 μm were trapped within the bags, but anything smaller or thinner could pass freely in or out.

### 2.1.6. Sampling of Bags

Sites were checked every 6 months by divers. Once every year, between the end of October and the beginning of November, transmitters were changed, and bags retrieved (one set of three bags from each pole). The time of year was chosen to disturb the cysts as little as possible; the weather was cold, and cyst bags could be picked up and transported to the laboratory without adding disturbance that could induce germination. After bags were taken up, permanent microscope mounts were made, and germination experiments were performed. The permanent cyst mounts were transported to Sweden for identification and counting of cysts. The retrieved bags were treated as follows: bags still attached to the rope were placed in a 1 L beaker with filtered seawater and sonicated for 5 min. Then, the beaker was placed on ice under black plastic, and each bag was treated separately by spraying thoroughly on the outside with filtered seawater using a plant sprayer. When the outside was clean, the bag was opened over a 100 μm screen standing on top of a 20 μm screen, and the contents were rinsed with filtered seawater. The resulting fraction was washed into a 100 mL volumetric flask, and volume was adjusted to 100 mL with filtered seawater. The samples were kept cold and dark and used for germination experiments, as well as making permanent microscope slides. Samples in test tubes were washed into a 100 mL flask with filtered seawater and treated as described for the samples from bags.

### 2.1.7. Permanent Microscope Mounts from Sampling

For each sample, the 100 mL flask was sonicated 2 min (from 2010: 5 min); then, an exact volume, containing enough sediment to make permanent microscope mounts, was taken and rinsed on a 20 μm screen with distilled water. The resulting fraction was washed into a centrifuge tube and placed in a refrigerator (4 °C, darkness) overnight to settle. The amount of sediment remaining was much less in the oxygenated environment compared with the anoxic environment and test tubes already in the first year. From the anoxic environment and test tubes in the refrigerator, 5.0 mL samples were taken; however, from the oxygenated environment, the volume needed for permanent mounts was larger (10.0 to 25.0 mL, differing between samples depending upon sediment content). After settling overnight, overlying water was removed with a pipette, and the remaining sediment was mixed. This material was divided into equal drops containing material of suitable density for permanent microscope mounts (for finding enough cysts, but not with material too dense). For each sample, at least four permanent microscope slides were made using the same method as for the starting samples (heating to 50 °C, adding to glycerin drop, and sealing with beeswax). For the mounts, the drop containing the sample was added directly to the glycerin drop, but drops only used for counting were placed on a separate microscope slide to confirm equality of drop size and material content, as well as for accurate counting. The number of drops was 10 to 23 (thus, the material in the slide constituted, for example, 1/10 or 1/23 of the total sample in the test tube, and could be calculated with reference to the content of the entire bag).

### 2.1.8. Germination Experiments

As described above, the material used for germination experiments was rinsed with filtered seawater to constitute only the 100–20 μm fraction and sonicated 5 + 2 min (from 2010, 5 + 5 min). The goal of the germination studies was to observe and identify dinoflagellates to group level in slurry cultures. Other emerging protists and organisms were identified to group level, and their presence was noted. Then, 50 mL Nunclon or Falcon tissue-culture flasks fitting to inverted microscopes were used because they permit the study of slurry cultures over time without subsampling and with little disturbance. Briefly, 50 mL of the

rinsed fraction was added to the flask, and the flask was filled to the neck with f/2 medium (~15 mL; flasks available in the laboratory were somewhat different between years but could be fitted to the inverted microscopes). This was to avoid air bubbles that obstruct the vision in the microscope and to add some nutrients. The flasks were placed in an incubator (16–20 °C, 14 h/10 h light at ~200 $\mu E \cdot m^2 \cdot s^{-1}$) with caps not screwed tight to allow gas exchange. Observations of dinoflagellates and other organisms were made daily in an inverted microscope (Olympus IX51, Olympus Corp., Tokyo, Japan, and Axio Observer Z1, Carl Zeiss microscope, Germany) for at least 4 days and 10 days at the most (2009: 5, 2010: 4, 2011: 5, 2012: 10, 2014; 10 days), thorough notes were made (with sketches/line drawings), and photos were taken when possible. Dinoflagellates swim quickly; without capture and the use of slowing agents or plate coloring, they can only be identified to group level. There are great advantages to slurry cultures because organisms are only slightly disturbed, and cultures can be studied over time. The objective with the highest magnification possible to use with 50 mL flasks was 40× (+ 10× in the eyepiece, resulting in 400× magnification). At this level of magnification, it is possible to identify dinoflagellates by their characteristic swimming motion, while their color and general shape are clear (autotrophic or heterotrophic, round, drop-shaped, apex shape, the shape of horns, large sulcal list, etc.); however, details (such as plate patterns) are not seen, and only a few species are so characteristic that identification to species level is possible. For a more thorough identification, in the last year (2014), the remaining samples were concentrated by filtration and viewed at high resolution. Extra slurry cultures were also made using the extra sediment stored in the same refrigerator (the concentrated cyst fraction saved from the start). Slowing of swimming and reduced destruction of living cells was induced by the addition of methylcellulose. Coloring of plates was applied using Calcofluor white to view the plate patterns. A Zeiss Axioskop 2 Mot Plus microscope (Germany) was used. All triplicate samples from each pole and test tube were germinated and followed by microscopy each year. Notes of species observations were made and pooled from the triplicates to yield results for each treatment and year.

### 2.1.9. Microscopy and Naming of Samples

Microscopy of permanent mounts was performed over several years because of the large number of samples and the time-consuming nature of the work; however, all samples were counted (in random order) in the same microscope (Sagitta L3000) and by the same person (A.P.). Three samples (permanent microscope slides) were counted entirely for each treatment and year (one from each pole and year, except for the oxygenated site where, from the year 2010, only one pole was found. Then, three samples from this same pole were counted. The last year, only two poles were found on the anoxic site, and then one sample from one pole and two samples from the other pole were counted). All cyst results are expressed as cysts per bag (originally 10.0 mL wet weight of concentrated cyst fraction, corresponding to 3.9 g dry weight). Dinoflagellate cysts studied were larger than 20 μm and trapped within bags, but material or organisms smaller than the mesh size were not controlled or studied. In the figures, samples denoted "oxygenated" are those placed in New Haven harbor, "anoxic" denotes bags placed in the borrow pit of Morris Cove, "test tubes" are samples in test tubes kept in the refrigerator, and "bag" denotes the three bags that were kept inside a jar with 1 L of original sediment sample in the refrigerator until 2014 (6 years unopened). "Extra sediment" in germination trials denotes saved, concentrated sediment stored since the start of the experiment.

### 2.2. Statistical Analyses

Statistical analyses and comparisons among treatments, species, and groups in the data were performed using R 4.0.3 (R Core Team (2021) R: A language and environment for statistical computing), MS Excel Professional Plus 2016 (version 2209) AnalysisToolPak, and StatGraphics Plus 5.1. (Statpoint Technologies Inc., Warrenton, VA, USA). For each dependent variable and point in time, comparisons were made using one-way ANOVA.

The method used to discriminate between means was Fisher's least significant difference procedure at the 95% level ($p < 0.05$). Error bars in figures represent the standard error of the mean.

## 3. Results

### 3.1. Cyst Numbers and Diversity

A large number of dinoflagellate resting cysts was found: $1.4 \times 10^4$ resting cysts·cm$^{-3}$ sediment ($3.6 \times 10^4$ resting cysts·g$^{-1}$ dry weight) in the concentrated cyst fraction. These were of over 62 different types, and 61% of the cysts were filled with live-appearing contents at the start of the experiment (Table S1). There was a large difference in preservation of cysts between the oxygenated and anoxic environment, and a large difference over time (Figures 2 and 3). In the oxygenated sediment, very few cysts remained after 1 year; both "living" (filled with live-like contents) and empty cysts disappeared. Filled cysts showed a 95% decrease, and empty walls of cysts decreased by 80% compared to start values. Furthermore, in anoxic sediment samples, the number of living cysts decreased significantly with time, but not quite as fast. After 1 year, 65% of the living cysts in the anoxic environment had disappeared (or become empty), and 30% of the living cyst refrigerated in test tubes had disappeared or become empty. After 4 years, 79% of the original number of cysts with contents in the anoxic environment had disappeared or become empty, and 69% of those in test tubes.

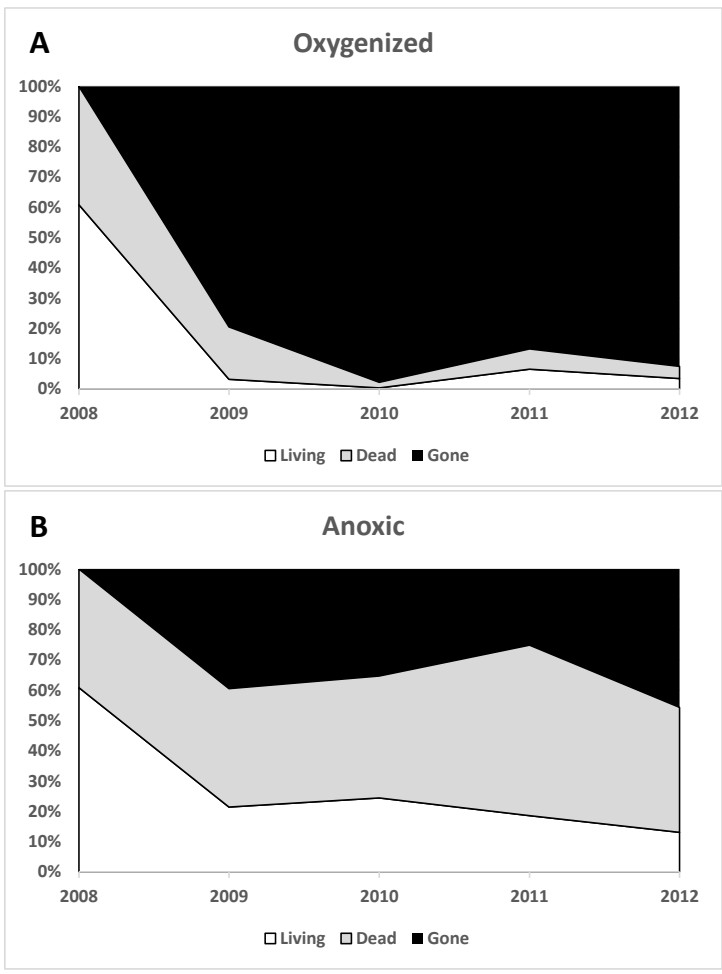

**Figure 2.** *Cont.*

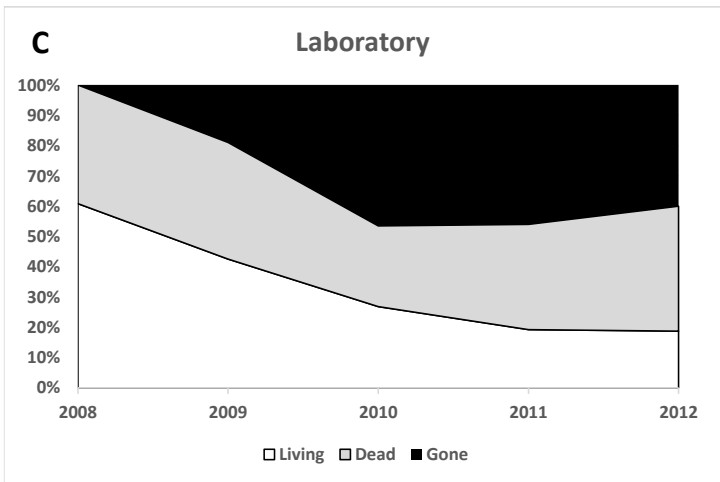

**Figure 2.** The average disappearance of cysts over time in the different environments during the first 4 years. The black area denotes disappeared cysts, gray represents dead and empty cysts, and white denotes filled live-appearing cysts. (**A**) oxygenized, (**B**) anoxic, and (**C**) laboratory (refrigerated) environments.

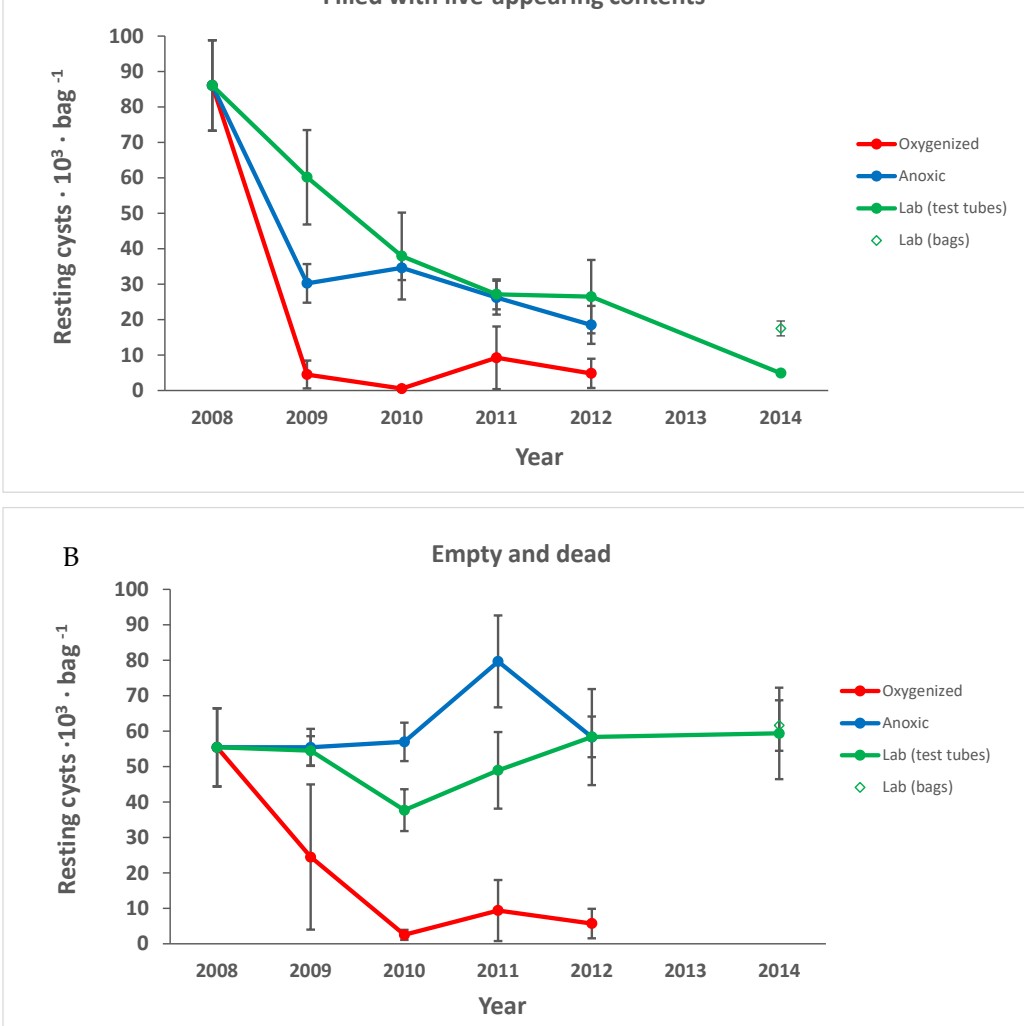

**Figure 3.** *Cont.*

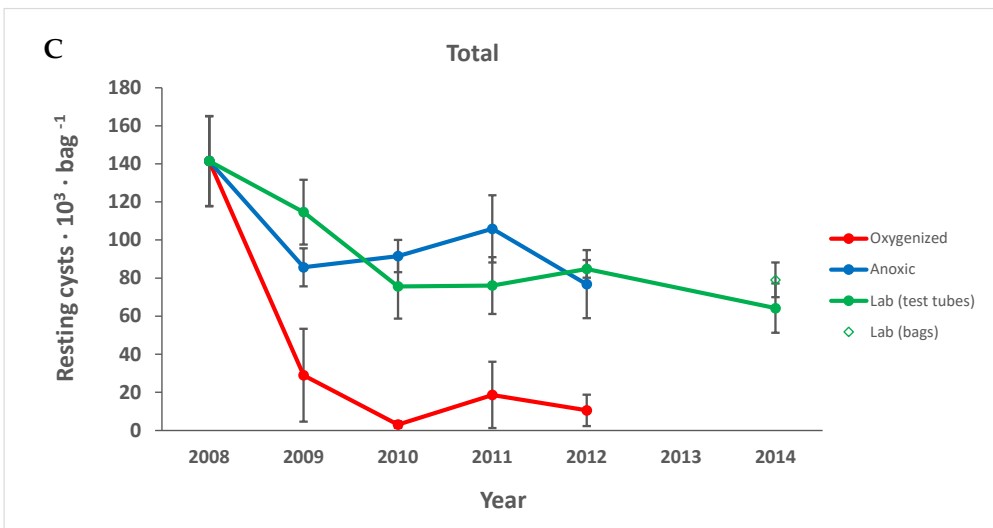

**Figure 3.** Cysts presumed to be alive, with live-like contents (**A**) decreased in all treatments, while the number of dead and empty cysts (**B**) decreased only in the oxygenated environment. The total number of cysts is shown in (**C**). Points are averages and error bars show the standard error of the mean.

### 3.1.1. Preservation of Empty Cyst Walls

The empty cyst walls remained intact for a long time in the anoxic environment and refrigerator, especially for many species known to fossilize well, but the number of empty cysts also decreased with time (after 4 years, 46% of total cysts had disappeared from samples in the anoxic environment, and 40% from those stored in the refrigerator).

### 3.1.2. Variation

Variation was quite large between replicates in the same treatment (Figure 3) even though the sediment was well mixed before concentration in the sorter, during sorting, and before adding the concentrated cyst fraction to the bags. The anoxic macroenvironment was homogenous throughout the experimental period. The bags seemed unaffected, had no fouling growth, and the sediment within the bags was black (Figure 4A). There was no significant difference in cyst numbers or species composition among the three poles in the anoxic environment; the variation found was between samples (bags). The variation was, thus, intrinsic within the original concentrated sample that was divided into bags at the start of the experiment. In the oxygenized macroenvironment, there was a large variation even within short distances; bags tied to the same rope, ca. 20 cm apart, sometimes had one bag seemingly unaffected and containing ample sediment, while the bag a short distance from it was overgrown with bryozoans and nearly empty (Figure 4B).

### 3.1.3. Macroenvironment Observations

An enormous difference between the anoxic and oxygenated macroenvironments was clear from observations during sample recovery in the field and from the observations on the bags as mentioned above (Figure 4). The anoxic environment contained dark, fluffy, anoxic sediment without animals. The oxygenized sediment was characterized by the presence of fauna and more coarse particles.

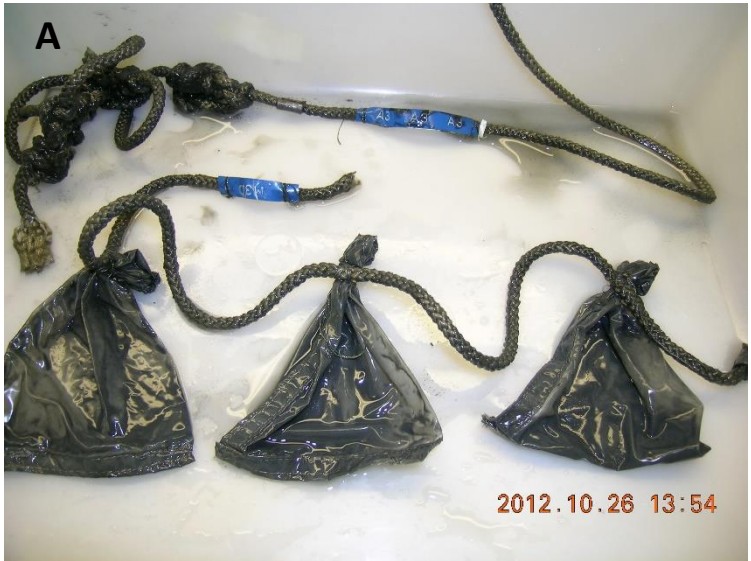

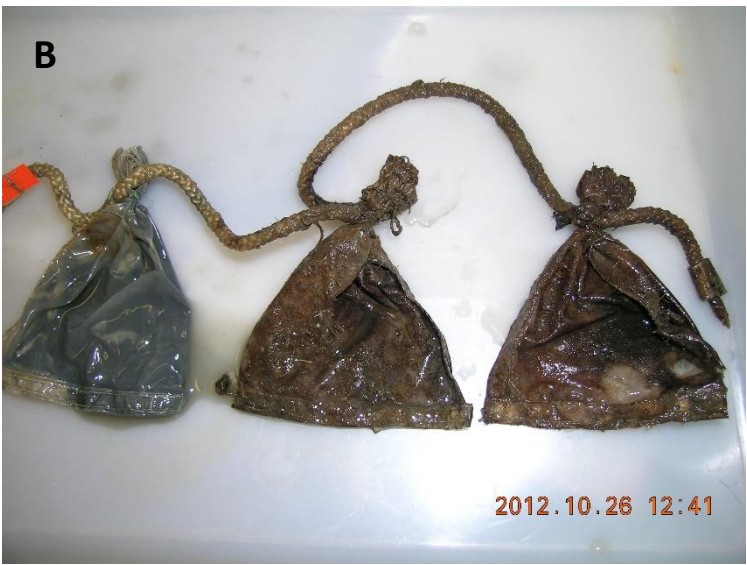

**Figure 4.** Bags from the anoxic (**A**) and oxidized (**B**) environments.

### 3.1.4. Species-Dependent Differences in Preservation

There were species-dependent differences in preservation; this is most easily described when dividing cysts into categories, because many species constituted a very small part of the total cyst assembly (Tables S1 and 1). They could be divided into the most common species and groups or according to their ability to fossilize (non-fossilizable and fossilizable). We use the categorization found in the literature regarding fossilizability and also introduced a third category, i.e., the round gray smooth cyst, since this was a very common cyst type. We suspect that it represents an undescribed *Kryptoperidinium* species that was very frequent in the slurry cultures.

**Table 1.** Resting cysts recorded in permanent cyst mounts and corresponding vegetative cells found in slurry cultures. Groups: U = unfossilizable, F = fossilizable, G = round gray smooth (*Kryptoperidinium* sp.?). The proportion is the percentage (± standard deviation) of the cyst assembly in the original concentrated sediment at start.

| Group | Proportion | Resting Cysts in Permanent Mounts | Vegetative Cells in Slurry Cultures |
|---|---|---|---|
| *Gonyaulacales* | | | |
| U | <1% | *Alexandrium* spp. | *Alexandrium* spp. |
| F | <1% | *Ataxiodinium choane* | Unknown, likely autotrophic |
| F | <1% | *Bitectatodinium tepikiense* | *Gonyaulax* sp. |
| U | <1% | *Gonyaulax verior* | *Gonyaulax verior* |
| F | <1% | *Lingulodinium polyedrum* | Not seen (has characteristic shape) |
| F | <1% | *Nematosphaeropsis labyrinthus* | *Gonyaulax spinifera* |
| F | 25% ± 3% | *Pentapharsodinium dalei* | *Pentapharsodinium dalei* |
| F | <1% | *Protoceratium reticulatum* | *Protoceratium reticulatum* |
| F | <1% | *Pyrophacus steinii* | Not seen (has characteristic shape) |
| F | <1% | *Spiniferites bentori* (=*Gonyaulax digitalis*) | *Gonyaulax digitalis* |
| F | <1% | *Spiniferites bulloideus* (=*Gonyaulax scrippsae*) | *Gonyaulax spinifera*/*Gonyaulax* sp. |
| F | <1% | *Spiniferites* cf. *furca* | *Gonyaulax spinifera* |
| F | <1% | *Spiniferites* cf. *membranaceum* | *Gonyaulax spinifera* |
| F | <1% | *Spiniferites mirabilis* | *Gonyaulax spinifera* |
| F | <1% | *Spiniferites elongatus* | *Gonyaulax spinifera* |
| F | <2% | *Spiniferites* spp. unidentified | *Gonyaulax spinifera* |
| *Gymnodiniales* | | | |
| F | <1% | *Gymnodinium microreticulatum*/*nolleri* spp. | *Gymnodinium* sp. |
| F | 4% ± 1% | *Polykrikos kofoidii* | *Polykrikos* |
| F | <1% | *Polykrikos* other spp. | *Polykrikos* |
| F | <1% | *Margalefodinium polykrikoides* | *Margalefodinium polykrikoides* |
| *Peridiniales* | | | |
| F | <2% | *Diplopelta symmetrica* | *Diplopsalis* group |
| F | <2% | *Diplopelta parva* | *Diplopsalis* group |
| F | <1% | *Diplopelta latipeltata* | *Diplopsalis* group |
| F | <2% | *Diplopsalis lenticula* | *Diplopsalis* group |
| F | <1% | *Diplopsalis orbicularis* | *Diplopsalis* group |
| F | <2% | *Diplopsalis* group, unidentified | *Diplopsalis* group |
| F | <1% | *Echinidinium* with thin processes | *Diplopsalis* group |
| F | <2% | *Echinidium* cf. *aceulatum* | *Diplopsalis* group |
| F | <1% | *Echinidinium*, spp. with capitate processes | *Diplopsalis* group |
| F | <1% | *Echinidinium*, spp. with acuminate processes | *Diplopsalis* group |
| F | 3% ± 1% | *Echinidinium*/*Islandinium* spp., other | *Diplopsalis* group |
| F | <2% | *Islandinium* sp. | *Diplopsalis* group |

**Table 1.** *Cont.*

| Group | Proportion | Resting Cysts in Permanent Mounts | Vegetative Cells in Slurry Cultures |
|---|---|---|---|
| F | <1% | *Operculadinium israelianum* cf. | *Diplopsalis* group |
| F | <1% | *Protoperidinium americanum* | Dinoshaped heterotroph |
| F | <1% | *Protoperidinium* cf. *americanum*, other spp. | Dinoshaped heterotroph |
| F | <1% | *Protoperidinium claudicans* | *Protoperidinium* sp. |
| F | <1% | *Protoperidinium stellatum* | *Protoperidinium* sp. with small antapical horns |
| F | <1% | *Protoperidinium conicoides* | Elongate rounded heterotroph; swimming wiggly |
| F | <2% | *Protoperidinium conicum* | *Protoperidinium* sp. with small antapical horns |
| F | <1% | *Protoperidinium leonis* | *Protoperidinium* sp. with small antapical horns |
| F | <1% | *Archaeperidinium minutum* | Dropshaped heterotroph, cf. *P. minutum* |
| F | <1% | *Protoperidinium nudum* | Dropshaped heterotroph, round with pointed apex |
| F | <1% | *Protoperidinium oblongum* group | *Protoperidinium* sp., large |
| F | <1% | *Protoperidinium pentagonum* | *Protoperidinium* sp. with small antapical horns |
| F | <1% | *Protoperidinium* spp. unidentified | *Protoperidinium* sp. |
| F | <1% | Square brown cyst with membrane | Unknown, likely *Protoperidinium* sp. |
| U | <1% | *Scrippsiella* cf. *crystallina* | *Scrippsiella* sp. |
| U | <1% | *Scrippsiella trochoidea* | *Scrippsiella* sp. |
| U | <3% | *Scrippsiella* spp. without crystals, elongate | *Scrippsiella* sp. |
| U | <2% | *Scrippsiella* sp., spherical | *Scrippsiella* sp. |
| U | <1% | *Scrippsiella* without and with dissolving crystals, spherical | *Scrippsiella* sp. |
| U | <1% | *Scrippsiella* spp. other | *Scrippsiella* sp. |
| | | **Unidentified dinoflagellate cysts** | |
| F | <1% | Brown with buds/dents/spikes, different spp. | Unknown, likely heterotrophic |
| F | <1% | Unidentified brown spp. | Unknown, likely heterotrophic |
| F | <1% | Colorless spiny spp. | Unknown, likely autotrophic |
| U | 3% ± 4% | Colorless spp. With rugged surface/mucus | Unknown, likely autotrophic |
| U | 4% ± 4% | Colorless smooth spp. | Many different autotrophic |
| F | 11% ± 2% | Round brown smooth spp. | Different Diplopsalis and Protoperidinium spp. |
| G | 15% ± 4% | Round gray smooth | *Kryptoperidinium* sp. ? |

All types of cysts decreased with time, but compared to non-fossilizable species, both fossilizable cyst walls and the gray cysts, were rather resistant in the anoxic environment and lab storage (Figure 5). All types of cysts decreased significantly within the first year in the oxygenated environment (Figure 5).

The decrease seen for fossilizable cysts in Figure 5 is, to a large extent, attributable to *Pentapharsodinium dalei*, which decreased in all treatments over time (Figure 6A). *Spiniferites* spp. that are also fossilizable decreased significantly only in the oxygenated treatment, and some other species within the group well-known to be very resistant fossilizable were only found empty, e.g., *L. polyedrum*, *P. reticulatum*, and *Pyrophacus steinii* (cyst name *Tuberculodinium vancampoae*). *Protoperidinium* spp. and resting cysts of the family *Diplopsalidaceae* decreased significantly only in the oxygenated environment (Figure 6B,C) while non-fossilizable cysts decreased with time in all treatments (Figure 5B). The gray smooth

cyst thought to be *Kryptoperidinium* sp. decreased significantly only in the oxygenated treatment (Figure 5C). *Scrippsiella* spp. were common in the group considered non-fossilizable and decreased with time in all treatments (Figure 7).

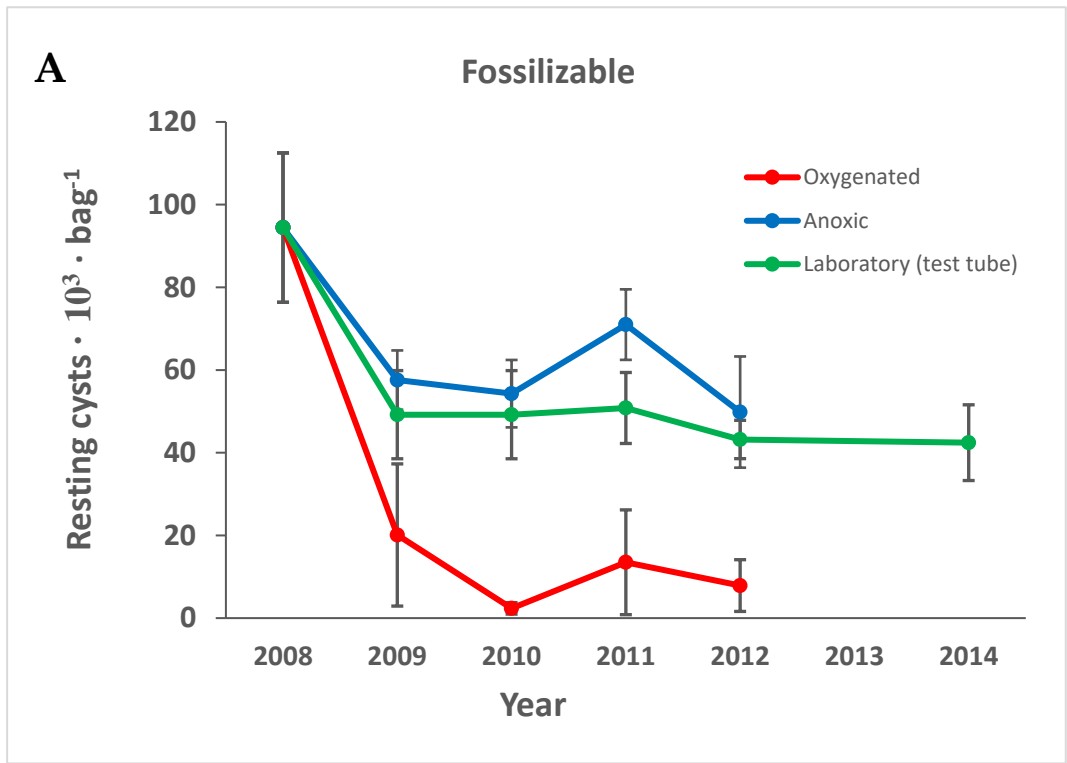

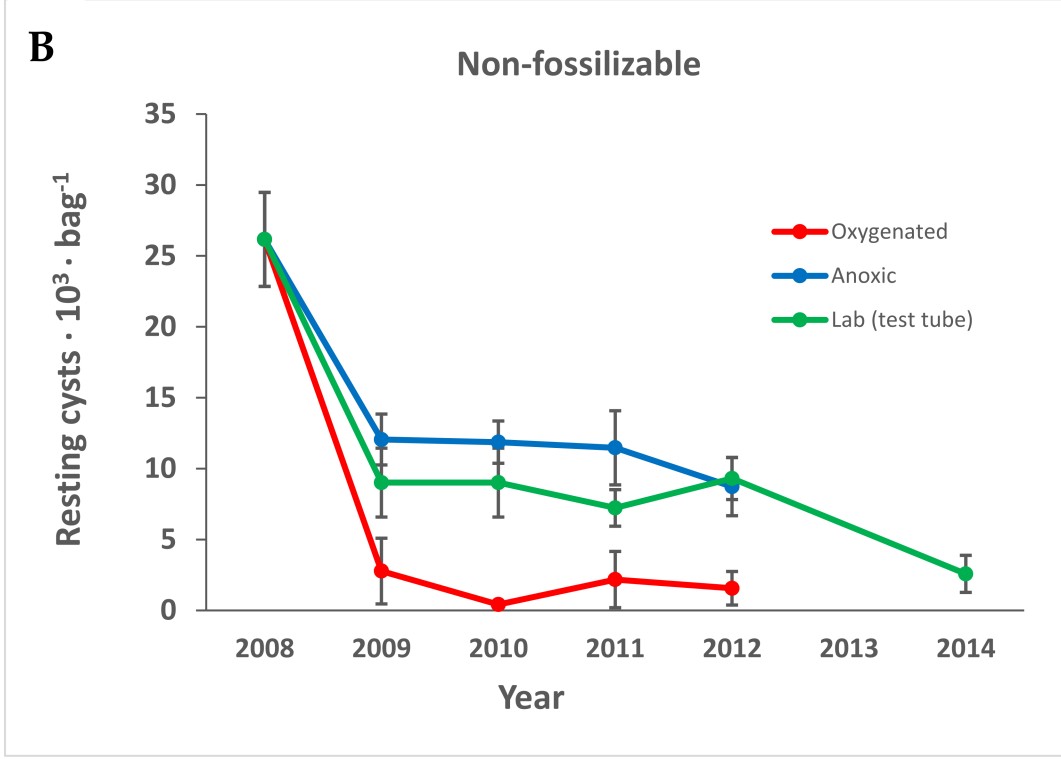

**Figure 5.** *Cont.*

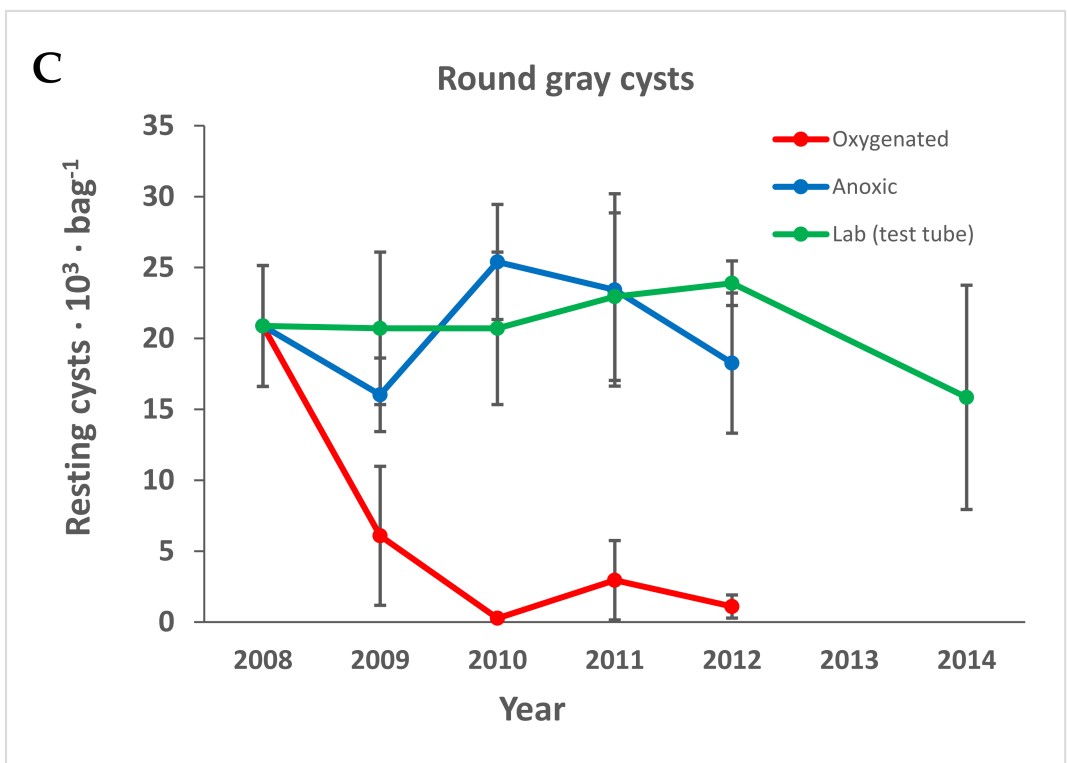

**Figure 5.** The resting cysts divided into groups depending on fossilizability according to the literature: (**A**) fossilizable; (**B**) non-fossilizable; (**C**) round gray cysts. Error bars are the standard error of the mean. Species are divided into groups as shown in Table 1.

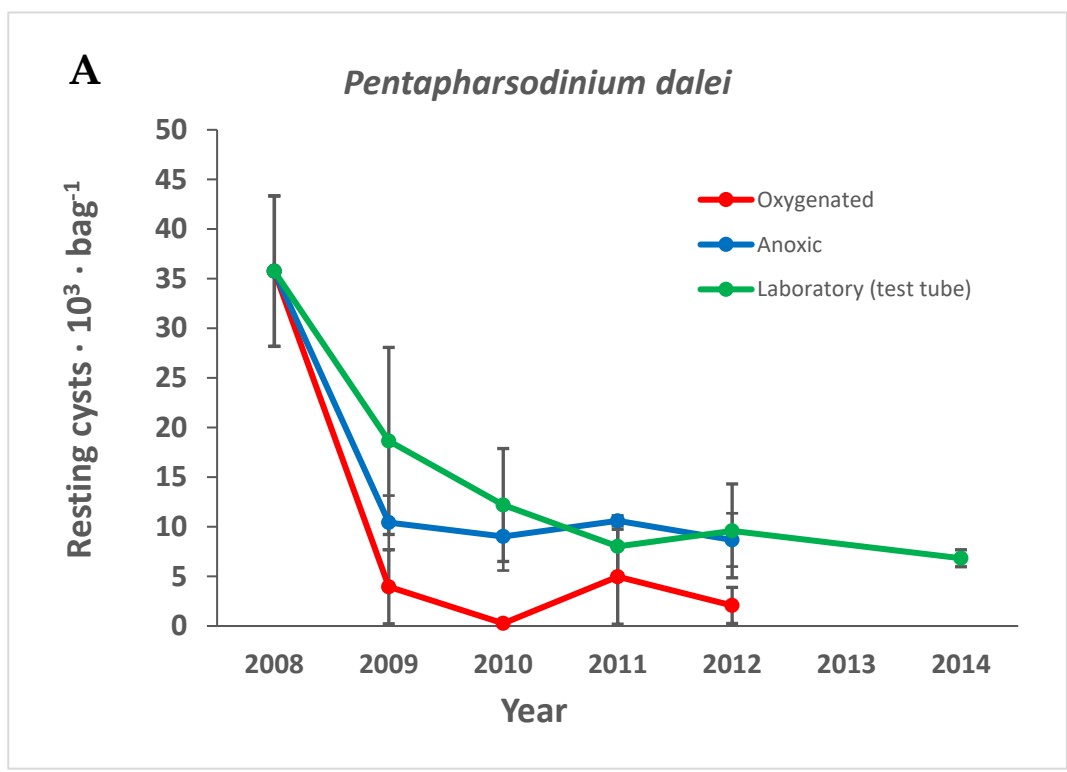

**Figure 6.** *Cont.*

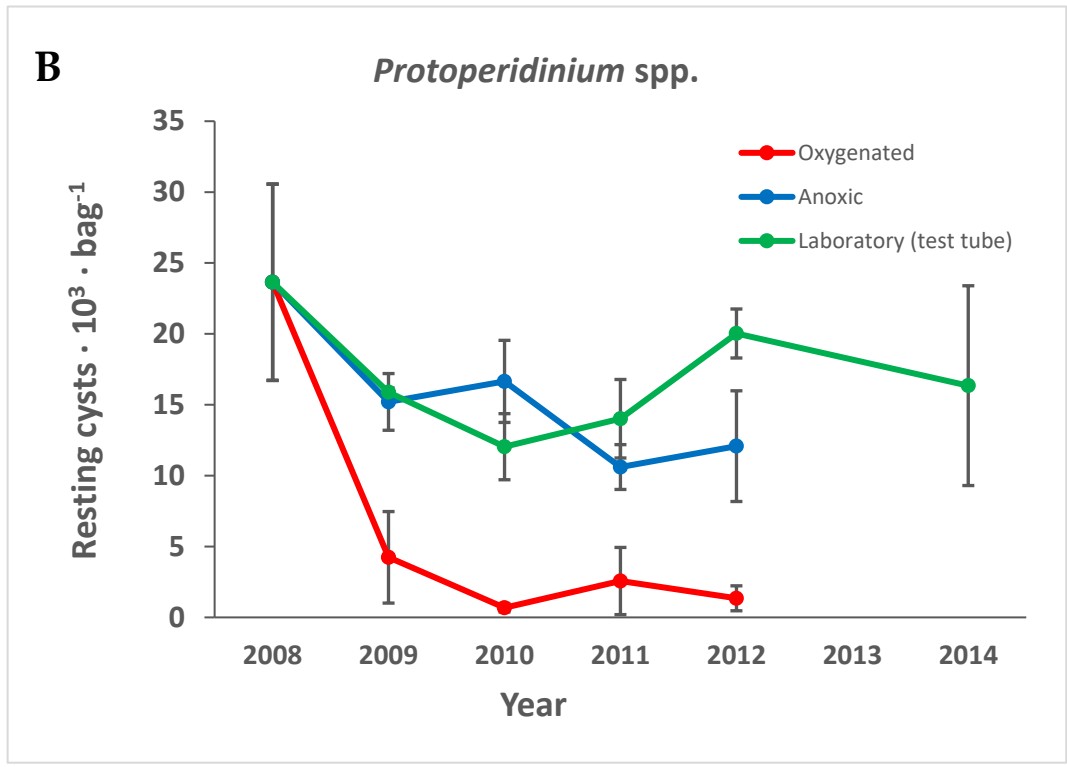

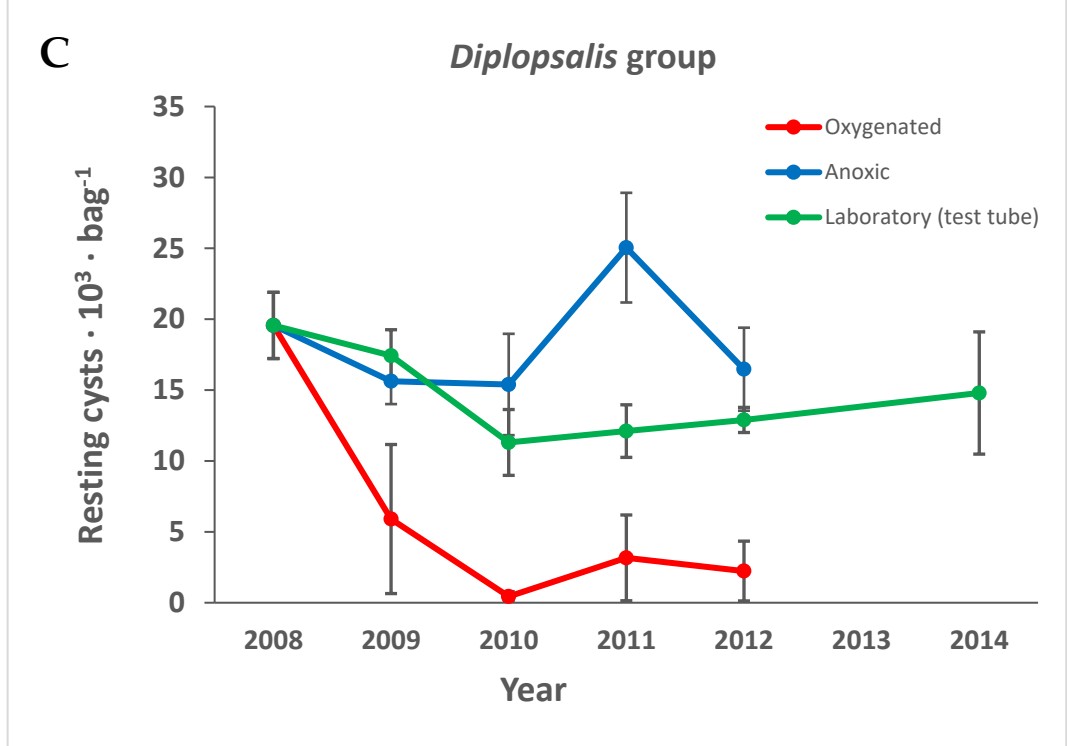

**Figure 6.** The number of (**A**) *Pentapharsodinium dalei*, (**B**) *Protoperidinium* spp., and (**C**) resting cysts of the *Diplopsalis* gruop over time in the different treatments. Error bars are standard error of the mean.

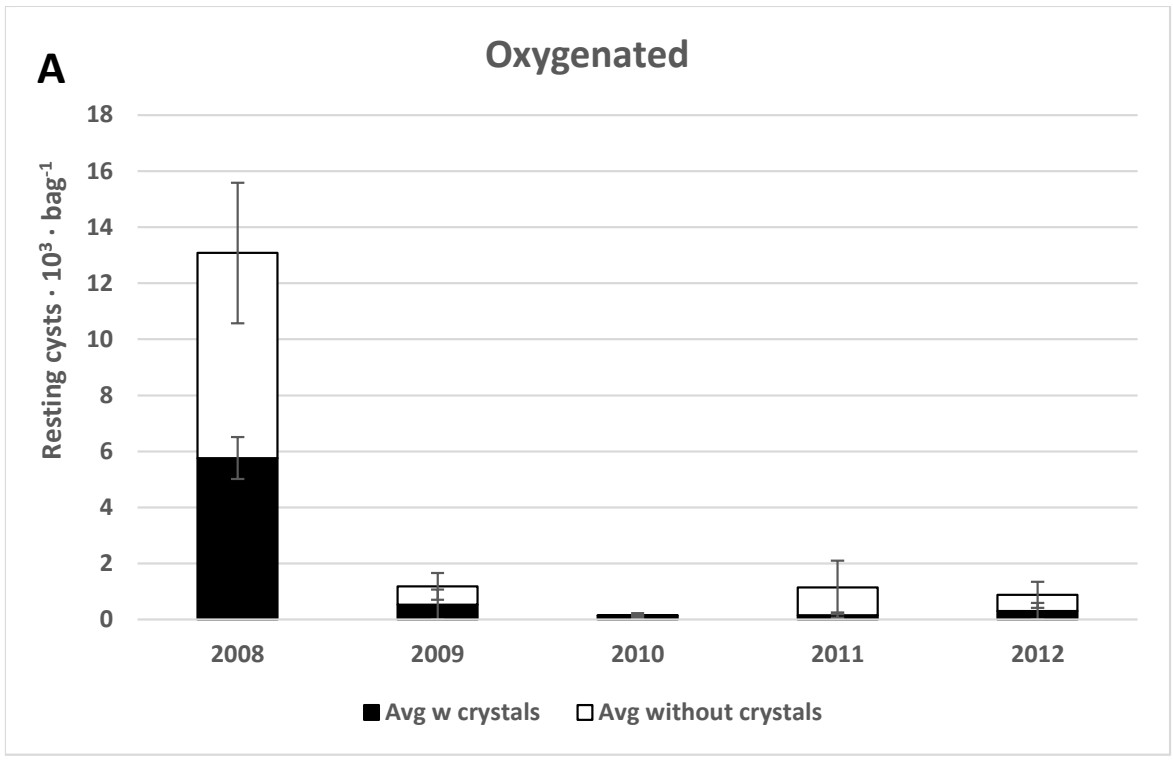

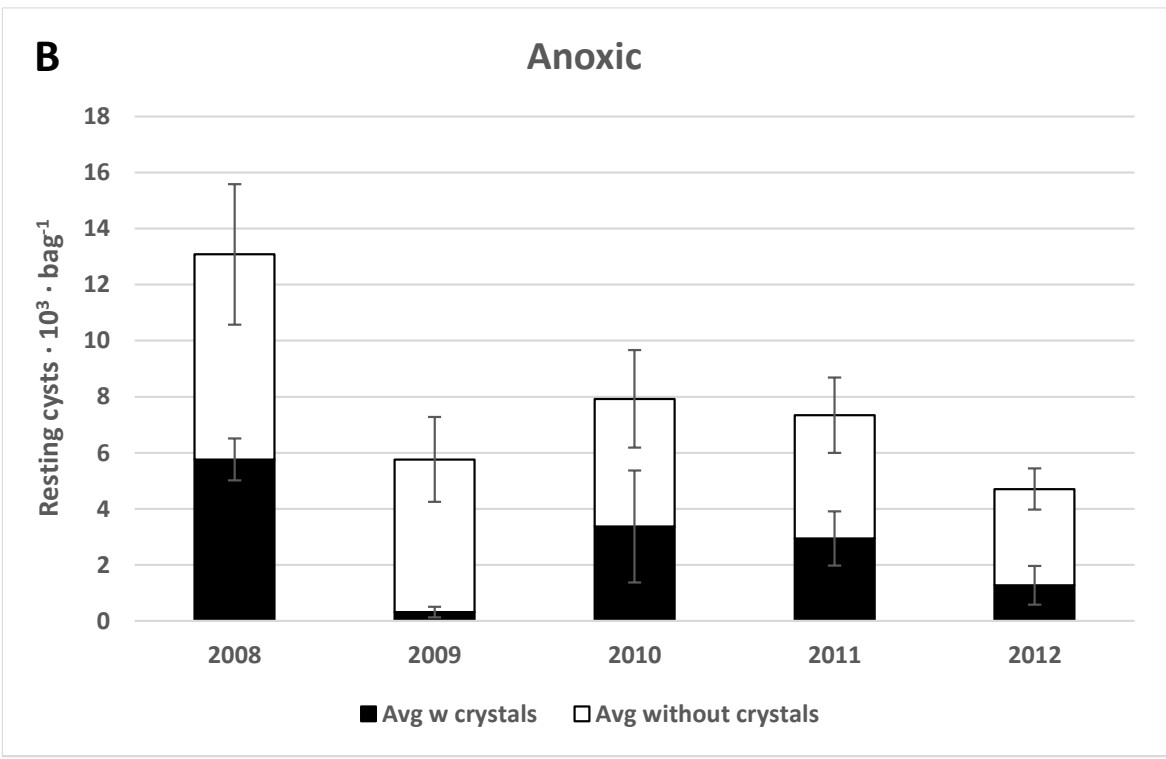

**Figure 7.** *Cont.*

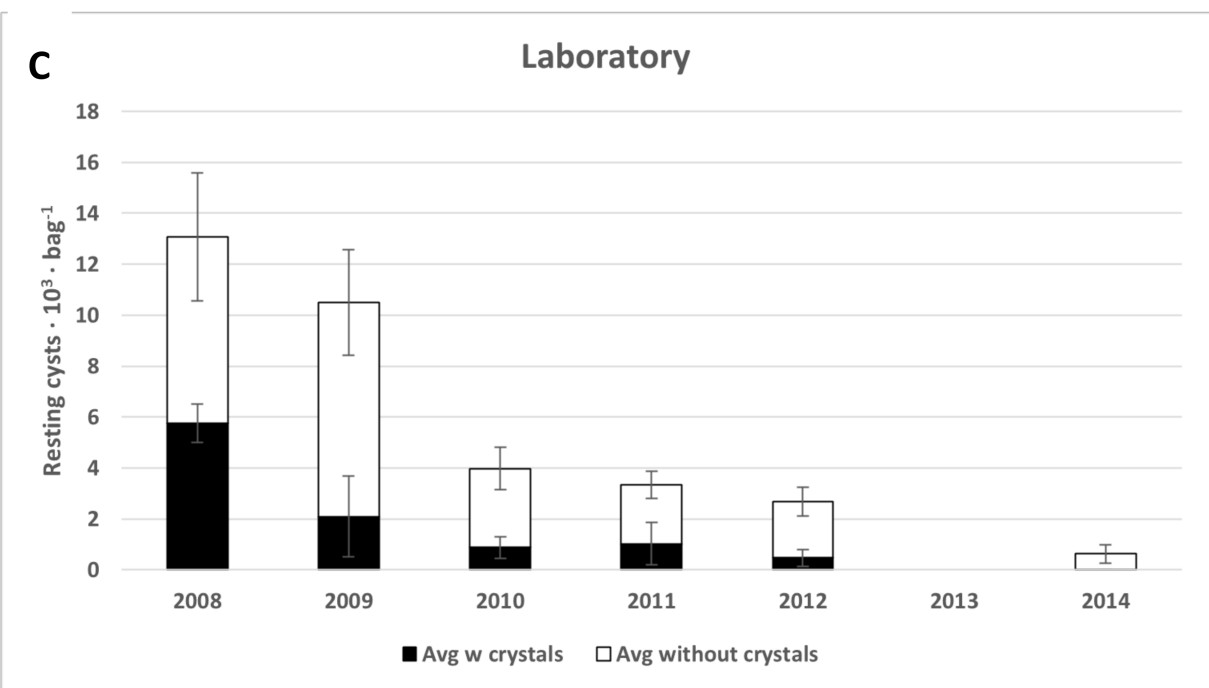

**Figure 7.** The total number of *Scrippsiella* spp. cysts with (black) and without (white) crystals. (**A**) oxygenized, (**B**) anoxic, and (**C**) refrigerated (test tube) environments. Error bars are the standard error of the mean.

For resting cysts in oxygenated sediment, there were no significant trends for species or groups. The majority of all cysts disappeared during the first year irrespective of fossilizability (Figures 5–7).

### 3.2. Gemination Experiments

Germination of sediment samples resulted in slurry cultures with a rich variety of dinoflagellate species. All species with identifiable cysts were found also as living, germinated cells (Table 1), except two of those that were found only empty (*L. polyedrum* and *P. steinii*). It is not possible to count cells in slurry cultures, but the number of morphological groups found was recorded (Table 2).

**Table 2.** The number of different morphological groups of vegetative cells observed in slurry cultures over time. "Bags" denotes the three bags that were kept inside a jar with sediment in the refrigerator until 2014 (6 years unopened). "Extra sediment" is the concentrated cyst fraction stored until 2014 in the same refrigerator.

| Number of Germinated Dinoflagellate Species/Groups in Slurry Cultures | 2009 | 2010 | 2011 | 2012 | 2014 |
|---|---|---|---|---|---|
| Morris cove (anoxic) | 13 | 17 | 13 | 10 | |
| New Haven Harbor (oxygenated) | 5 | 3 | | 6 | |
| Laboratory (test tubes) | 14 | 16 | 19 | 1 | 1 |
| Laboratory, extra sediment (anoxic) | | | | | 19 |
| Laboratory, bags in sediment (anoxic) | | | | | 12 |

As in other germination experiments using slurry culturing [44,55], dinoflagellates with unknown cysts appeared. The most common species was a dorsoventrally flattened, thecate, heterotrophic dinoflagellate with a prominent and characteristic red eyespot (Figure 8A). A round gray-to-light-brown cyst with a smooth surface, differing from other round brown dinoflagellate cysts by the much lighter and more grayish color (Figure 8D),

was very numerous in the cyst preparations; therefore, we tentatively propose that it belongs to this species. Other dinoflagellates appearing in slurry cultures without being found as cysts are species with unknown cysts, e.g., *Amylax triacantha*, *Fragilidium* sp., *Gonyaulax* cf. *polygramma*, *Heterocapsa rotundata* (cf.), *Peridinium quinquecorne*, *Protoperidinium* cf. *divergens*, *Nematodinium armatum*, and several small dinoflagellates (autotrophic and heterotrophic) whose cysts must have remained despite rinsing on the 20 μm sieve.

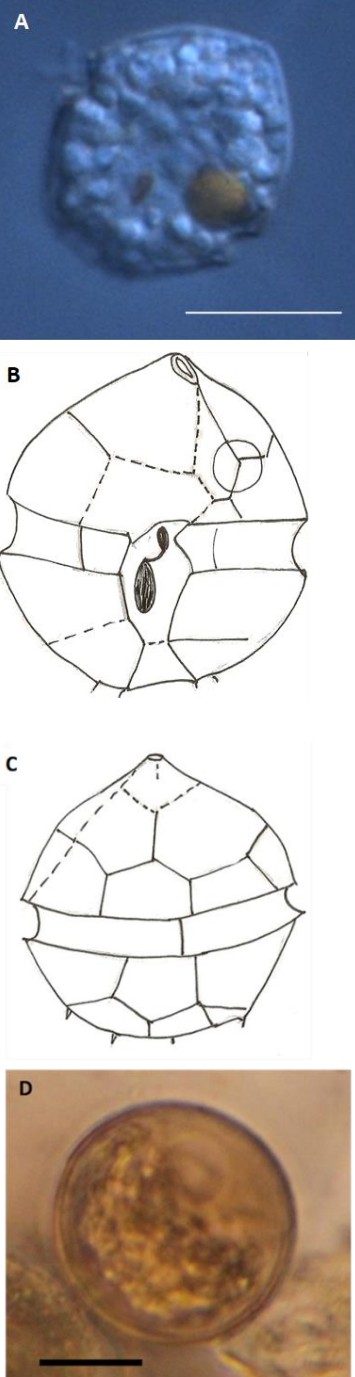

**Figure 8.** *Kryptoperidinium* sp. (**A**) The dorsoventrally compressed cell was colorless apart from a characteristic bright-red eyespot. A large vacuole, shifting in placement, often had red-brown contents. (**B**,**C**) Suggested plate pattern from Calcofluor-colored cells. Broken lines are uncertain; the very small spikes were often more than four and placed in a half-circle. (**D**) The smooth gray cyst was common (possibly *Kryptoperidinium* sp.?) The scale bar is 10 μm.

Refrigerated samples in polypropylene test tubes did not germinate after 4 years, even though plenty of cells had germinated from this treatment in all previous years (Table 2). Many dinoflagellates germinated from refrigerated sediment after 6 years when stored in a jar with a large amount of sediment (either in bags within anoxic sediment: "bags") or from the remaining original concentrated sediment.

### 3.3. Major Groups and Species of Specific Interest

#### 3.3.1. Kryptoperidinium sp.

A previously undescribed flat, heterotrophic species with a characteristic eyespot was very common in the incubations (slurry cultures), as mentioned above. It germinated quickly (within 2 days), in every slurry culture, and persisted throughout the entire observation periods. When studied more in detail, both plate pattern and eyespot had a resemblance to *Kryptoperidinium foliaceum* (Figure 8). This species was never, however, encountered with chloroplasts and had no chlorophyll fluorescence. It always had a bright-red eyespot. An unidentified, smooth, round, gray to light-brown cyst was very common in the cyst assemblage (Figure 8D), and we suspect that it belongs to this species, even though no single cell incubations were performed. This cyst decreased in numbers only in the oxygenated treatment (Figure 5C).

#### 3.3.2. Diplopsalidaceae

Diverse living (filled), smooth, as well as spiny, brown cysts were found (Tables S1 and 1, Figure 9A–D). Round, brown, smooth cysts usually belong to *Diplopsalidaceae* or *Protoperidinaceae*, but spiny brown cysts usually belong to *Diplopsalidaceae*. In slurry cultures, many slightly different living vegetative cells of the *Diplopsalidaceae* were found. They germinated quickly (within 1–2 days) and were seen in all cultures but were not possible to identify to species level in slurry cultures through the inverted microscope. Cyst numbers of *Diplopsalidaceae* decreased only in the oxygenated treatment (Figure 6C).

#### 3.3.3. *Scrippsiella* spp.

Resting cysts of several different *Scrippsiella* species were common in the original concentrated sample (Tables S1 and 1, Figures 7 and 9E–H), and 90% of them were filled with live-appearing contents at the start of the experiment. Vegetative cells were often observed in germination experiments. More than half (56%) of the cysts were without calcite crystals, and both cysts with and without calcite crystals decreased significantly with time during the experiment (Figure 7). The bright-red accumulation body often remained in empty cysts (Figure 9H).

#### 3.3.4. *Pentapharsodinium dalei*

*P. dalei* resting cysts (Figure 9I) were very common and constituted a large part of the original concentrated cyst sample (25% $\pm$ 3%, Table 1). The vegetative cells were observed frequently in slurry cultures. The cyst numbers decreased significantly with time in all treatments (Figure 6A).

#### 3.3.5. Gonyaulacaceae

Many different vegetative cells of the family *Gonyaulacaceae* were observed in the slurry cultures. Some vegetative cells are very characteristic and could be identified in low-resolution microscopy (e.g., *Peridinium quinquecorne*, Table 1), but others could be identified to group level only (Table 1). Some species in the group that germinated had unknown cysts (e.g., *Amylax triachantha*), some had well-known but unfossilizable cysts (e.g., *Gonyaulax verior*, Tables S1 and 1), some had well-known cysts of questionable preservability (e.g., *Alexandrium* spp. Tables S1 and 1), and others had well-known resistant fossilizable cysts (e.g., *Spiniferites* spp. Tables S1 and 1). Many different *Spiniferites* resting cysts have paleontological species names, but all germinate into similar vegetative cells (*Gonyaulax spinifera*, Table 1), and this is regarded as a species complex.

*Peridinium quiquecorne* germinated in the slurry cultures, even after 6 years of storage (Figure 9L). It was not found as an intact cell in cyst preparations (as it was found in high numbers from a different location in Long Island's Peconic Estuary; East Creek).

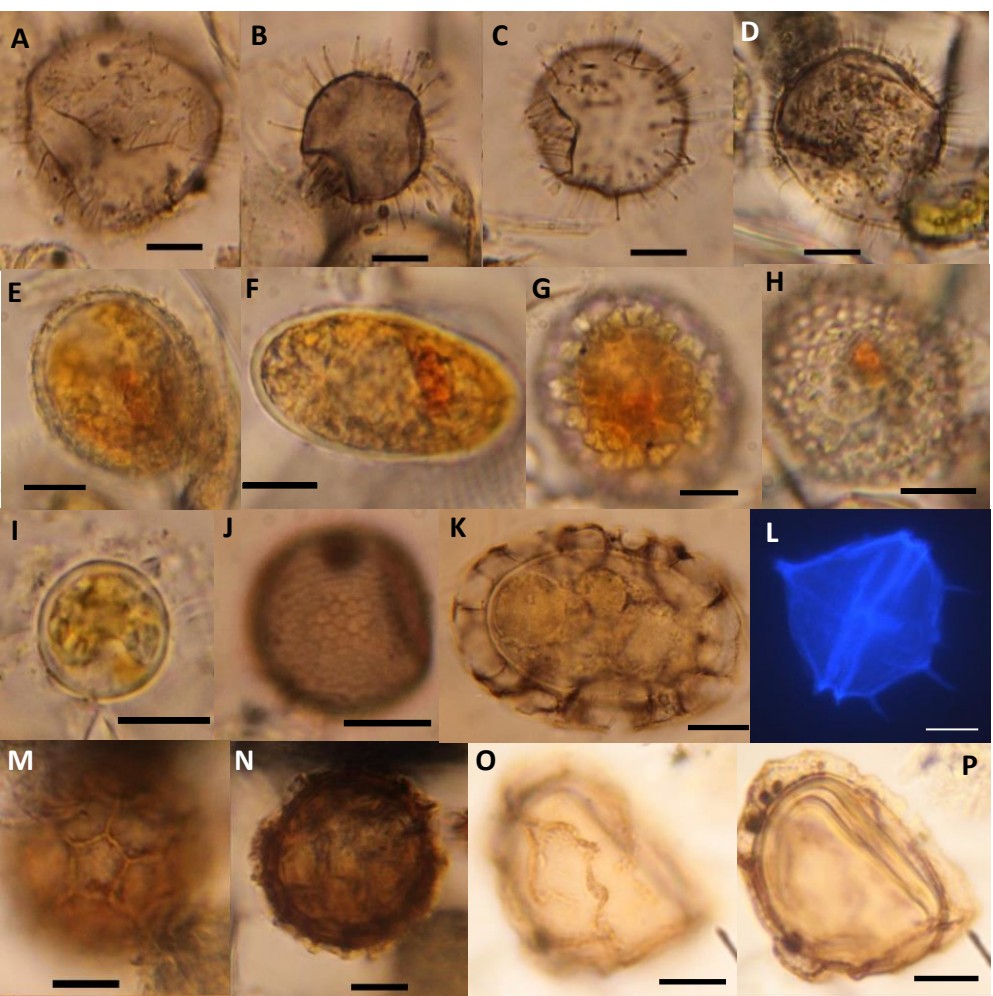

**Figure 9.** (**A–D**) Four examples of different spiny brown resting cysts. (**E–G**) *Scrippsiella* sp. resting cysts with and without crystals. (**H**) Empty *Scrippsiella* sp. resting cyst containing red spot. (**I**) *Pentapharsodinium dalei* resting cyst. (**J**) *Gymnodinium microreticulatum* resting cyst. (**K**) *Polykrikos kofoidii* resting cyst. (**L**) *Peridinium quinquecorne* vegetative cell colored with Calcofluor. (**M–P**) Two types of *Margalefodinium polykrikoides* resting cysts. (**M**) Type 1 surface, and (**N**) cross section. (**O**) Type 2 surface, and (**P**) cross section. The scale bars represent 10 μm.

### 3.3.6. Protoperidinaceae

Many *Protoperidinium* species with well-known, fossilizable cysts were found and identified (Tables S1 and 1); others were smooth round brown cysts that could not be identified to the species level when alive (they are identified by the shape of the archeopyle: the opening where the cell exits at germination). Vegetative cells of *Protoperidinium* are sometimes characteristic with one prominent apical horn and two antapical horns, while others are very similar to the *Diplopsalidaceae*. Many *Protoperidinium* spp. were seen in slurry cultures (Table 1).

### 3.3.7. Gymnodiniales

Microreticulated round brown cysts of the group *Gymnoidinium microreticulatum/nolleri* were found in low numbers (Tables S1 and 1, Figure 9J).

*Polykrikos kofoidii* cysts were common, both filled and empty (Tables S1 and 1, Figure 9K).

*Margalefodinium polykrikoides* cysts of two types were found in low numbers, both filled and empty (Figure 9M–P). *M. polykrikoides* germinated in slurry cultures and were seen as vegetative cells.

### 3.4. Macrofauna

Macrofauna recorded in the original sediment was sparse and near the surface of the mud. Organisms observed (from the total 76 L of mud) were as follows: one northern moon snail, *Lunatia heros*; 20 soft shell clams, *Mya arenaria*; 12 winkles, *Littorina Littorea*; two polychaetes, *Nereis* sp.

### 3.5. Meiofauna

Meiofauna observed in the germination experiments were nematodes, copepods, ciliates, cladocerans, and rotifers. Copepod, rotifer, and ciliate eggs were found in the sediment samples, but not counted. Nematodes were numerous in all environments after 1 year, and disappeared from the test tubes after 2 years, but were present after 6 years in anoxic samples stored in the same refrigerator. They were plentiful initially in slurry cultures but declined in numbers during the incubation period.

## 4. Discussion

### 4.1. Preservation under Anoxic versus Oxygenated Conditions

In this study, we present the first quantitative measurements on the preservation of living dinoflagellate resting cysts in different environments in nature (to our knowledge). The vast differences apparent in preservation under anoxic versus oxygenated conditions point to the magnitude of the environmental changes induced when the extent of hypoxic zones on the seafloor increases. Hypoxic zones are reservoirs of resting stages and of slowly decomposing organic material.

The results are independent of grazing effects, as the bags excluded macro-grazers. The additional effect of grazing must, however, be included in the discussion, as animals in a healthy, oxygenated seafloor environment exert a large influence on the degradation of organic material, including protistan resting stages. Under hypoxic conditions, the nutrient web with micro-, meio-, and macro-fauna as recyclers of nutrients in material originating from sedimenting primary producers is disrupted. Resting stages of harmful organisms, such as toxin-producing dinoflagellates, are preserved in anoxic environments, together with nutrient-rich organic material. When this material (harmful organism and nutrients for growth) is brought to surface waters through upwelling or manmade construction actions, there is an increased risk of harmful blooms and "vicious cycles" of nutrient webs without organisms that can be harvested for consumption or sustain a healthy aquatic environment.

### 4.2. Natural Variation in Dinoflagellate Cyst Beds

4.2.1. Variation between Seasons

The literature describes extensive variation in sediment dinoflagellate cyst amounts and compositions between seasons, followed over one or several years [56–61]. Lee et al. [59] found that the abundance of resting cysts varied extensively by month. Anderson et al. [56] found that, by June 1980, new cysts constituted 65–95% of the total cysts in sediment samples from three studied salt ponds in Cape Cod. Anderson et al. [57] described that the area of seabed locations varied 3–4-fold and total cyst abundance varied more than 10-fold across years. Erard-Le Denn et al. [58] found extensive variation by months in abundance and species composition of dinoflagellate cysts in sediments. Gao et al. [61] found differences in the cyst bank between seasons. Diaz et al. [60] found that *A. catanella* cysts disappeared quickly from the sediment; after 3 months, less than 10% remained. We interpret this well-documented variation between seasons as reflecting not only cyst production, germination, and burial but also consumption by grazers causing death and disintegration of cyst walls into small fragments through feeding actions such as grinding and chewing. The difference between oxygenated and anoxic environments in the ability

to act as seed (cyst) banks is well known. Miyazono et al. [62] found that the viability of *A. tamarense* cysts decreased rapidly in aerobic but slowly in anaerobic conditions. Anderson et al. [63] described a method for the storage of samples and stated that anoxia is effective in maintaining quiescence without excessive mortality. Dinoflagellate cysts have a long survival potential in anoxic sediments. The literature mentions the germination of cysts after a long time (40 years: [20]; 60 years: [64]; 100 years: [62,65]). Lundholm et al. [20] described that, when samples were exposed to oxygen, the ability to germinate was reduced. In the present experiment, several groups of dinoflagellates still germinated after 6 years, and there is no reason to doubt that cysts may stay alive for a very long time under anoxic conditions. It is, however, also important to note the considerable decrease in numbers of living cysts over time, as also seen in anoxic environments. It is relevant to talk about a "half-life" of cysts (as discussed in [66,67]), different for each species and certainly different when it concerns living cysts or empty walls. Keafer et al. [66] concluded that the half-lives of living, buried cysts were in the range of years to decades, and Lewis et al. [67] suggested that half-lives of 2–10 years might be expected for living, buried cysts. Some cysts have walls that preserve very well and make up a fossil record that remains intact over a geological timescale of many millions of years, but other cysts do not preserve and are destroyed by palynological preparation methods.

### 4.2.2. How to Study the Same Living Sediment Sample over Time

No previous cyst studies have permitted the comparison of the same original living sample over several years in nature. For that, an enclosure is necessary. Kodrans-Nsiah et al. [46] and Gray et al. [47] placed natural cyst communities in dialysis bags in different environments in nature and found that the species-selective degradation of dinoflagellate cysts in oxygenated natural environments is a rapid process. In their studies, palynological methods were used, and no living cysts or unfossilizable cyst species were studied. Their results nonetheless revealed that early diagenesis takes place rapidly under aerobic conditions and cannot be neglected on any timescale, whereas no degradation could be detected under anaerobic conditions during the study periods of 15 months [46] or five years [47].

### 4.2.3. Variation within Sites or Samples

Counting and identifying dinoflagellate cysts is very time-consuming. The overwhelming majority of all cyst studies (both surveys and sediment core studies) are un-replicated, leaving the variation within sites unknown. Our study provides unique insight into the intrinsic variation within a cyst sample. Our original sample was well mixed in several steps before being placed in bags. The same person (A.P.) counted all cysts, and samples were counted in random order. The variation seen between replicates was large considering the mixing, cyst numbers, and the species composition. Microclimatic differences between bags could occur over short distances in the field, but only in the oxygenated environment where macrofauna could move the bags, bury them, and lift them up. Even if the top layer is well oxygenated and rich in fauna, in an organic-rich environment, the sediment is often anoxic a short distance down, and a buried bag could be exposed to anoxic conditions. The anoxic environment in Morris Cove did not offer variation between bags attached to the same pole because no fauna was present, and the environment was very homogeneous. No difference was found between poles. Instead, the variation was found to be between bags within the environment. The sediment added to each bag at the start of the experiment was not mixed to a microscopic level. It was not sonicated. Sonication is used in dinoflagellate cyst studies as the cysts attach to debris, and they have spines, patterns, or a sticky mucus layer. Dirt surrounding cysts and clumping obstructs the view, and cleaning is necessary. During encystment, zygotes attach to marine snow, many at the same time of the same species (zygotes form by mating during a bloom); therefore, cysts often occur in clumps of the same species. Before counting and identification, samples are sonicated, sieved, and, in paleontological studies, treated with strong acids. It is not possible to know if cysts

were in a clump before this treatment. In the experiment described here, large clumps of cells were excluded because they sink faster and end up too far down in the sorter to be included in the fraction used. This applies, for example, to copepod fecal pellets, which were very common in the original sample, but not in the dinoflagellate cyst fraction used. The variation seen despite mixing in several steps shows that small differences in nature should not be over-interpreted, especially not in oxygenated conditions where large differences in the environment exist within short distances.

### 4.3. Oxygen Availability and Organic Matter Degradation

### 4.3.1. Microbial Decomposition of Sediment

For accurate measurements of changes in cyst numbers over time, an experimental approach enclosing a certain volume of sediment containing cysts is necessary. In the laboratory, it is possible to feed a certain amount of sediment containing cysts [24] or laboratory-cultured cysts [22,41] to animals in a container of known volume. From a natural sediment sample, not only cysts disappear, but the sediment itself becomes decomposed, which creates problems when quantitative comparisons are desired. A certain volume one year does not represent the same volume next year.

A large fraction of the newly produced carbon is rapidly recycled, and only a tiny fraction of the organic carbon export production reaches the seafloor [27,68]. Material deposited on the seafloor is further degraded by aerobic and anaerobic remineralization processes, and only a small fraction of the deposited organic carbon escapes benthic degradation [27,68–71]. Degradability is not an inherent or absolute property of the organic matter itself, but results from the interaction between the organic matter and its environment [27]. Benthic degradation of organic matter involves chemical, biological, and physical processes, and the microbial community exerts an important influence on its degradability, resulting from the combined effort of billions of individual microorganisms [27]. The anaerobic consortia of microorganisms are less efficient degraders of organic matter than aerobic microorganisms [68]. The disproportionate importance of aerobic microorganisms stems from their relatively high cellular activity, having 10–100 times higher respiration rate than sulfate-reducing prokaryotes deeper down in the sediment [27]. Aerobic prokaryotes can often live anaerobically and subsist until conditions improve, while many anaerobic prokaryotes are strict anaerobes and confined to the reducing subsurface sediment with low redox potential [27]. During degradation, cell-wall polymers (such as sporopollenin or the wall of fossilizable dinoflagellate cysts) are preserved well compared to polysaccharides, proteins, and other biopolymers [68]. Ultimately, the small faction of organic matter escaping the biochemical cycle into the geosphere is called kerogen [68] and is the origin of petroleum and natural gas.

### 4.3.2. Bioturbation and Grazing

Grazing and bioturbation happen at a fast rate at oxygenated sites. Filter feeders such as shellfish and meiofauna such as copepods and rotifers consume cysts before they even fall to the sediment such that many live and empty cysts occur within fecal pellets on the seafloor [72]. On the seafloor in aerobic sediments, deposit feeders often "rework" the upper centimeters of the sediment (e.g., polychaetes, holothurians, and bivalves) by feeding and burrowing, with one feeding on the fecal pellets of the other. The sediment is intensely reworked, as is everything in it. Moreover, very small grazers are common: microfauna and meiofauna. Every particle that sinks is consumed multiple times in the presence of oxygen. Different areas have a differently intense reworking of the sediment depending on the fauna present. Temperature, light, and changing redox conditions are important factors that influence remineralization processes. It is also known that short-term redox oscillations promote the degradation of organic matter [27,70,71].

The amount and species composition of dinoflagellate cysts found in an area, thus, does not exclusively reflect surface water productivity but is significantly altered after deposition. It is difficult to separate the effects of different factors such as grazing, selective

aerobic degradation, differences in upper water column conditions, wind-transported material, and sediment accumulation rate, as well as winnowing, lateral transport and advection, and water depth [69].

### 4.4. Dinoflagellates as Part of the Micro- and Meiofauna

Microfauna are animals smaller than 100 μm or bacteria and protozoa [73]. Microbenthic activity stimulates degradation of refractory organic compounds [74]. The definition of meiofauna varies, but generally denotes mobile, multicellular animals (non-protozoans) that are smaller than macrofauna and larger than microfauna. Size boundaries are based upon the standardized mesh apertures of sieves [75]. Characterized by high abundance and diversity, widespread distribution, and rapid generation times, meiofauna are important contributors to ecosystem processes and functions, including nutrient cycling and provision of food to higher trophic levels [42,43]. Oxygen consumption rates in meiofaunal organisms are greater than in macrofaunal organisms ([42] and references therein). Over the most recent four decades, the extent of hypoxic zones has increased globally, and dramatic decreases in meiofaunal abundance and biomass have been detected in association with hypoxia and anoxia [42,76]. In this experiment, meio- and microfauna were present within the bags as resting stages or could enter and leave the bags if they were smaller or thinner than 20 μm. Copepods, rotifers, cladocerans, and ciliates were commonly observed in germination experiments, and dinoflagellates may also consume other dinoflagellates when hetero- or mixotrophic. Many ciliates, rotifers, and cladocerans, along with at least 54 species of marine calanoid copepods, are known to produce resting stages or resting eggs (review in [77]; list and references in [78]).

During the experiment, only the oxygenated treatment presented conditions enabling germination of resting stages. Meiofauna within the bags, including dinoflagellates, may have germinated during the growth period and consumed what was possible before dying and decomposing. Germinated dinoflagellates were trapped within the bag and did not have access to the free water or enough light, neither could they escape germinated predators. Predators thinner than 20 μm, such as nematodes, could enter the bags to consume dead cells and microorganisms, perhaps even cysts. In the anoxic environment, germination of meio- and microfauna, as well as dinoflagellates and other phytoplankton, was prevented by anoxia, even when temperatures rose during summer months. In refrigerated samples, oxygen became available slowly through pores in the plastic, but conditions were cold and dark, and germination was inhibited. Slowly increasing oxygen availability may have increased cyst metabolism, preparing for germination that never happened, and after 4 years, exhausted all germination possibilities. Binder and Anderson [79] investigated the respiration of *Scrippsiella trochoidea* cysts as newly formed, quiescent, and before germination. They found increased respiratory activity before germination. In contrast, quiescent cysts displayed greatly reduced metabolic activity. Dinoflagellate cysts germinated after 6 years when kept in bags in the same refrigerator as the test tubes. These samples were stored inside anoxic sediment in the 4 L flask containing 1 L of the original unsorted sediment. Likewise, cysts from the remainder of the concentrated cyst fraction from the start of the experiment germinated after 6 years in the same refrigerator. Nematodes seemed to thrive in the anoxic refrigerated environment, but not in test tubes after 2 years. They were lively and plentiful at the start of slurry cultures, but not observed as frequently later on. Their identity, activity, and feeding mode while in the cold and dark storage conditions remain unknown. Nematodes are described to have a high anaerobic capacity [43,80], as well as a daily ingestion rate equal to their bodyweight [80]. Heip et al. [80] described free-living nematodes as the most important animal taxon in all marine sediments and mentioned a species living in apparently oxygen-free environments that cannot survive normal oxygen conditions. The aim of our study was to compare the preservation of dinoflagellate cysts over time in different environments. All animals and their activities must be considered as integrated parts of each environment.

Heterotrophic and mixotrophic dinoflagellates graze, often on organisms larger than themselves, overlapping with copepods [81,82]. The abundance of heterotrophic dinoflagellates in marine coastal environments may be equivalent to or greater than that of ciliates [81]. Heterotrophic dinoflagellates occur in many sizes, from picoplanktonic smaller than 3 μm [83] to several 100 μm. The free-living small dinoflagellates are many, both auto- and heterotrophic, but are largely unstudied. They germinate in slurry cultures of untreated sediment samples from different areas of the world (personal observations from Sweden, USA, and Israel), as well as from sediment stored for a long time (i.e., they have resting cysts). It is time-consuming to sieve these cysts; moreover, they are too small to study under a light microscope, as are their germinated cells. The difficulty to study them results in scant literature but does not prove that they are unimportant in nature. Dinophyceae is a large and important group of protists that are not only primary producers and bloom formers but also constitute an important part of the microfauna.

### 4.5. Preservability of Living and Dead Dinoflagellate Cysts

There is a great difference between the preservability of the contents of cysts and the walls of cysts. In a geological sense, living versus empty cysts is not an issue; all are dead after enough time, but empty walls provide information. The same is true for surveys; empty cyst walls found in an area show that the species most probably were present in the area even if they were not found in the plankton (hidden flora/fauna). From an ecological point of view, however, it matters very much if, for example, cysts of toxic species stay alive for a longer time because of the spread on anoxic sea bottoms, if dredging leads to toxic blooms, or if most cysts are consumed by bottom fauna (absence of cyst beds despite recurring, toxic blooms).

Nature contains an abundance of resting stages. One can almost say that all small organisms are everywhere within their environmental limits [44,45,84]. It is clear from other studies that it is not the species with the most numerous resting stages that cause the largest blooms, even if these germinate. Conditions for growth after germination determine what proliferates in the water. Small organisms constitute the base of the food web; most become eaten, and few encounter favorable conditions for growth. Fenchel and Finlay [45] concluded that the species found at sampling are a function of habitat properties, not historical factors. Martin et al. [85] found that there was no relationship between the abundance of overwintering cysts and the magnitude of *Alexandrium fundyense* blooms, indicating that habitat properties determine blooms, not the number of cysts in the underlying sediment before the bloom. There is, however, a connection between anoxic seabeds and dinoflagellate blooms because sedimenting blooms consume oxygen and may cause anoxia [4]. Resting cysts cannot be consumed in the absence of fauna. Storms or upwelling can reinoculate the water mass with resting cysts from nutrient-rich anoxic seabeds. Accordingly, the presence of cysts does not predict an imminent bloom, but the spread of anoxic sea bottoms most likely increases the abundance of resting stages, as well as the risk of blooms to come, with species-specific preferences, environmental conditions, and coincidence determining what organisms proliferate at each time and place.

Fossilizable cysts are defined as those with walls withstanding palynological standard treatments [5,35,37,38,86]. They show species-specific sensitivity to treatment and preservability in nature [47,48]. The chemical composition of dinoflagellate cyst walls has been shown to be species-specific and differ between autotrophic and heterotrophic species [6–8]. During the time frame of this experiment, most fossilizable cyst walls were very resistant in the anoxic environment, as would be expected. The gray cysts thought to be the undescribed heterotrophic *Kryptoperidinium* species were also resistant and, in this regard, seem similar to other round brown cysts. In the oxygenated environment, fossilizable cysts were destroyed, as were unfossilizable cysts. Fossilizable cyst types known to be very highly resistant, constituted a small proportion in this experiment. Some were found only as empty, e.g., *L. polyedrum*, *P. steinii*, and *P. reticulatum*. All these have vegetative cells of characteristic appearance, identifiable as living cells in slurry cultures. *P. reticulatum*

vegetative cells were found (Table 1), but not *L. polyedrum* or *P. steinii*. The contents of a permanent cyst slide represent much less sediment than a slurry culture; compared to the seafloor, this reveals only that the species are or have been present in the area. Many fossilizable cyst walls may have been old already before the experiment as part of the background sediment comparable with inert material, such as mineral particles.

### 4.6. Major Species/Groups and Species of Special Interest

#### 4.6.1. *Protoperidinum*, Diplopsalids and Round Brown Cysts

The genus *Protoperidinium* Bergh and the Diplopsalids represent dominant heterotrophic groups. Some cysts are easily identified, but others are simply round, and termed "round brown cysts". These cysts cannot be identified alive; their most important identifying character is the archaeopyle shape, i.e., the opening through which the cell germinates [5,87,88]. Round brown cysts constituted an important part of the cyst assembly (Tables S1 and 1), and many different *Protoperidinium* and Diplopsaloid species germinated in slurry cultures (Table 1).

Diplopsalids constitute a large group of very similar vegetative cells: round colorless dinoflagellates with a short apical horn and a clearly visible sulcal list and wing. Some cysts have very different morphologies, belonging to different cyst-based genera, but others are very similar to each other. Some have a rugged surface and are difficult or impossible to identify to species level. Others are round with different types of spines (and colored from gray to light and dark brown). Round brown spiny cysts within the paleontological cyst genera *Echinidinum* and *Islandinum* can be very similar, and most do not have a description of the vegetative stage. Those with such a description describe cells similar to the Diplopsalids [89]. *Protoperidinium minutum* has a complicated taxonomic history and is more closely related to the Diplopsalidaceae than to other *Protoperidinium* species [90]. They are similar in appearance to Diplopsalids and represent a complex of species with very similar thecae but at least four different cyst morphologies [90–94]. Members of this group germinate within a short time in incubation experiments [90]. *Protoperidinium americanum* is also similar to the Diplopsalids when viewed as swimming cells at 400× resolution (slurry cultures), but the cyst is characteristic [94]. Similar cysts are described from other areas (e.g., *P. parthenopes* in Japan; [95]). Single-cell germination and identification with DNA methods are necessary for species-level determinations of *Dipolopsalis*-like cells [96]. Cysts within this group often germinate quickly in incubations (within 1–2 days; [90,93]), as was also observed here. Spiny round brown cysts were described in [86], and [93,97–99].

#### 4.6.2. *Pentapharsodinium dalei*

*P. dalei* is commonly regarded as a very resistant microfossil [10,100,101], yet it decreased in number over time in our study. *P. dalei* is sometimes used in paleoclimatic reconstructions [101] and is known for long survival times in anoxic sediments [102]. The present study is, to our knowledge, the first to quantitatively compare the preservation of recent living and empty *P.dalei* resting cysts over time and between treatments. Lundholm et al. [103] described decreasing viability with cyst age in a germination study where viable-looking cysts of different ages were picked for germination trials. Zonneveld et al. [48] recently showed that *P. dalei* is more sensitive compared to other fossizable dinoflagellate cyst species.

#### 4.6.3. *Scrippsiella* spp.

The genus *Scrippsiella* comprises several ubiquitous and cosmopolitan species, of which *Scrippsiella trochoidea* is probably the most well known [9,104,105]. This genus is widespread in coastal areas and in Long Island Sound (personal observations). *Scippsiella* resting cysts possesses an outer layer of calcite crystals [67,106], beneath which is a thin dinosporin wall. The inner wall is rather fragile and does not persist well in prolonged storage [67]. Calcitic cysts are most often not included in paleontological studies because methods for cyst preparation include removal of organic material by strong acids, which

also dissolves calcium carbonate [35]. Head et al. [33] found that the organic remains of *Scrippsiella trifida* cysts had been widely misidentified as *Alexandrium tamarense* (which is toxic and whose cysts have a low potential for geological preservation and are destroyed by standard palynological treatments). *Scrippsiella* spp. cysts survive without the outer crystal layer for some time. They can be found alive in sediment samples and germinate well in culturing studies [31,32]. Shin et al. [31] reported dissolution of calcareous spines of *S. trochoidea* cysts at pH < 7.39, indicating that loss of cysts is caused by low pH in acidic sediments of the hypoxic zone.

*Scrippsiella* spp. constituted several different species in this study. We previously found large variations in crystal structure within the same clone in cyst formation studies, as well as loss of crystals in grazing studies. The cysts used in this study may have been exposed to grazing before settling, and they were likely of different ages (and thus differently exposed to low pH). We found 56% *Scrippsiella* cysts without crystals in the original concentrated sample, and both cysts with and without crystals decreased significantly with time in all treatments. The degree of dissolution of crystals may have been both species-specific and time-dependent.

### 4.6.4. The *Gonyaulax* Group

The family Gonycaulacaceae is a large, ecologically important, and common group of autotrophic species, including several species with unfossilizable cysts (e.g., colorless and smooth; e.g., *Gonyaulax verior*), those with unknown cysts (e.g., *Amylax triacantha*, *Peridinium quinquecorne*), and many well-known microfossils, i.e., very resistant fossilizable ones such as *Lingulodinium polyedrum*, *Protoceratium reticulatum*, and the *Spiniferites* group of well-known and beautiful resting cysts [107]. Many cysts from the *Gonyaulax* group were present in the permanent cyst mounts (Table S1), and corresponding vegetative cells were found in slurry cultures (Table 1). The many different *Spiniferites* spp. cysts germinate into very similar cells: the *Gonyaulax spinifera* group [107].

### 4.6.5. *Margalefodinium polykrikoides*

We found *M. polykrikoides* resting cysts of two different morphological types (Figure 9M–P). One type (M–N) was as described in [108–110], while the other type (O–P) was described in [19,111]. Vegetative cells of *M. polykrikoides* were observed in slurry cultures. This species causes fish-killing blooms globally [12,112] and has a known presence in Long Island Sound [113,114]. It regularly causes ichthyotoxic blooms on the US east coast [113,114]. Blooms can be massive [115] but are also toxic at low cell density ([113]: *M. polykrikoides* ichthyotoxic at 3.3 cells·mL$^{-1}$). Jung et al. [115] described massive blooms despite a low density of the cysts and stated that the patterns of *M. polykrikoides* cyst abundance cannot represent prior blooming and might not predict future blooms.

### 4.6.6. *Peridinium quinquecorne*

*P. quinquecorne* is a well-known bloom-former with global distribution preferring eutrophic waters, where it causes fish mortality [116]. In an unpublished multiyear experiment using sediment from a different location in Long Island's Peconic Estuary (East Creek; 40°56′39.0″ N, 72°34′14.9″ W), many *P. quinqucorne* cells germinated. From this sediment sample, permanent mounts of cysts were made as described here. In those mounts, seemingly intact vegetative cells were found with live-looking contents. This is highly unusual for dinoflagellates, as repeated sonication and rinsing in distilled water destroy vegetative dinoflagellate cells. These cells probably represented some kind of temporary resting stage; however, they were not equal to the usual round type that represents the protoplast of a cell (such pellicle cysts form from different life stages, when disturbed, and can germinate after a day if conditions are good. They are better protected against e.g., grazing and coloring, but do not survive cyst preparation and mounting). The sediment was stored in a refrigerator for 3 months (4 °C). After one more year, many *P. quinqucorne* vegetative cells still germinated from the same sample in slurry cultures, and we searched

for and found the resting cells using density-gradient centrifugation with Percoll. As no single-cell germination trials were performed, we do not know if the resting cells germinated or if true resting cysts were also present. Other scientists have reported germination of *P. quinqucorne* from sediment samples [96,117,118]. Satta et al. [118] reported the same type of resting cells of *P. quinqucorne*, but no true resting cysts have yet been found or identified to our knowledge.

The findings here suggest there exists a true resting cyst because the species germinated in slurry cultures after 6 years of cold and dark storage, and no resting cells similar to the vegetative cells were found in the many permanent cyst mounts. The resting cyst could be of simple morphology, perhaps round and colorless and likely not fossilizable. This is a species of global interest that is possible to germinate from sediment samples. A morphologically distinct resting cyst likely would have been discovered and described earlier. *P. quinqucorne* is a successful bloom-former in East Creek; the sampled sediment was shallow and oxygenized, unlike Morris Cove, with its dark anoxic environment of accumulating fine-grained material.

### 4.6.7. *Kryptoperidinium* sp.

The most common dinoflagellate in slurry cultures was one that cannot be found in the literature: a dorsoventrally flattened colorless dinoflagellate with a bright-red eyespot. It germinated quickly and was numerous. A closer look at high magnification and coloring of plates with Calcofluor white revealed similarity with *Kryptoperidinium foliaceum*, in terms of cell shape, plate pattern, and morphology of the eyespot (Figure 8). However, it was never, at any instance, observed with chloroplasts or having chlorophyll fluorescence. It is without doubt a heterotrophic species. It has previously been observed in the area (pers. comm. Gary Wikfors), but not named. The group of species belonging to the Kryptoperidinaceae is interesting, and descriptions of new species are increasing ([119–121] and references therein). Many of them have diatom chloroplasts [122–126] but not all [121]. Kleptoplastidity has been shown for *Durinskia capensis* [127]. The dinoflagellates within the group have a characteristic eyespot [18,128,129], but some species and strains are without it [130]. There are also *Glenodinium* species described from freshwater that resemble the species we found [131].

Dodge and Crawford [128] and Kreimer [129] described the eyespot of *K. foliaceum*. The species common in our slurry cultures had an identical eyespot. The resting cyst assumed to belong to this species was smooth round and grayish to light brown (Figure 8D). It was very common in the samples, and different from other round, brown cysts by its distinctly more grayish color. Heterotrophic dinoflagellates often have brown cysts, and the *Kryptoperidinium* sp. cells were so common and germinated so quickly that their cysts must have been numerous in the cyst mounts. The number of cells found implies that this species may be important in the area as part of the microfauna, perhaps living near the sediment surface.

## 5. Conclusions

This study demonstrated great differences in dinoflagellate cyst preservation under oxygenated versus anoxic conditions. Under oxygenated conditions, cysts were quickly destroyed, regardless of their potential preservability as fossils, even when large grazers were excluded. Only 5% of the living cysts remained after 1 year without the presence of macrofauna. Adding the effects of constant grazing that occurs under normal oxygenated conditions, we can better appreciate the magnitude of differences that are propagated when the area of hypoxic zones increases. The spread of hypoxic zones on the seafloor contributes to the buildup of an increasingly larger reservoir of resting stages and enhances the risk of harmful blooms when cysts become able to reinoculate the water in favorable growth conditions. The presence of cysts does not predict an imminent bloom, but the spread of anoxic sea bottoms most likely increases the abundance of all resting stages ("good" and "bad"), as well as the risk of blooms to come, while environmental conditions and

coincidence determine what organisms succeed to proliferate at each time and place. There is a need to limit favorable growth conditions for harmful algae, which are often the very same that cause the spread of hypoxic zones: human-induced eutrophication.

**Supplementary Materials:** The following supporting information can be downloaded at: https://www.mdpi.com/article/10.3390/phycology2040022/s1; Table S1: Dinoflagellate resting cyst species.

**Author Contributions:** Experimental design, equal contribution; B.C.S. was in charge of fieldwork and diving, as well as practical moments during cyst sorting, access to laboratory materials, etc.; A.P. was responsible for culturing, microscopy, data acquisition, writing, analysis, and interpretation. All authors have read and agreed to the published version of the manuscript.

**Funding:** A.P. received travel grants from The Swedish Royal Scientific Academy (KVA), CF Lundströms Stiftelse grant No. 1390, and grants from The Ocar and Lili Lamm Foundation.

**Informed Consent Statement:** Not applicable.

**Data Availability Statement:** The data supporting the findings of this study are available within the article and its Supplementary Materials.

**Acknowledgments:** We wish to thank helpful reviewers and Gary Wikfors for constructive comments on the manuscript, Tom Barrell and Glenn Sharland of the "Seahawk", State of Connecticut, Aquaculture Division, for assistance on sampling and deployment of cysts bags and poles, respectively, and Robert Alix of the R/V Victor Loosanoff, NOAA/NEFSC, for subsequent yearly sampling occasions. Alice Stratton assisted Barry Smith in diving during sampling, and Mark S. Dixon and Ronald Goldberg were divers on the deployment of bags, with Barry Smith as the designated person in charge. Mark S. Dixon, David Villeaux, and Barry Smith were divers during yearly samplings.

**Conflicts of Interest:** The authors declare no conflict of interest.

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
