# Peer review of "Preservation of Dinoflagellate Cysts in Different Oxygen Regimes: Differences in Cyst Survival between Oxic and Anoxic Natural Environments"

_phycology, doi:10.3390/phycology2040022_

Round 1

Reviewer 1 Report

Review of Persson and Smith

This paper presents the results of a long-term experiment that tested the hypothesis that preservation of dinoflagellate cysts differs in aerobic versus anaerobic sediments. The data convincingly demonstrates that cyst viability, durability and fossilisation potential is significantly higher in anoxic sediments. The paper makes a unique and important contribution to the literature on the ecology of dinoflagellate resting cysts and is recommended for publication

The paper is very well written, largely error free and presents an impressive overview of the existing literature on this topic.    If anything, the paper is probably a bit too long, the literature citations too exhaustive and it contains a degree of repetition within the results and discussion sections that could be eliminated.  These are only minor quibbles however and do not detract from the merits of this manuscript.

A few more detailed comments follow:

·         In Figs 3,5,8 it would be helpful to have legends on the graphs showing the meaning of the coloured data points and lines rather than in the captions.

·         In Fig. 3 there is no line between anoxic treatment data points for 2012 and 2014. Although the data for 2013 may be missing it would still be possible to have a line and explain in the caption.

·         In Fig. 5,  C and D categories  are referred to in the caption but do not appear on the figures?

·         There should be a legend in Fig. 5.

·         In Fig. 7, category  C (refrigerated) referred to in the caption isn’t present as a figure?

·         There are no lines on the y axes in fig. 8

·         The use of literature citations in the results section (e.g. lines:  401-402, 457, 536, 553-554, ) is not good practise and these (along with the text associated with them) should be in the methods or discussion sections.

·         There seems to be quite a bit of overlap between sections 3.3 and 4.8.

·         Shouldn’t the genera and species names in the headings in section 4.8 be italicised?

·         Describing toxic species as “evil” (lines 401-402) is taking anthropomorphism a bit far

Author Response

Point 1: This paper presents the results of a long-term experiment that tested the hypothesis that preservation of dinoflagellate cysts differs in aerobic versus anaerobic sediments. The data convincingly demonstrates that cyst viability, durability and fossilisation potential is significantly higher in anoxic sediments. The paper makes a unique and important contribution to the literature on the ecology of dinoflagellate resting cysts and is recommended for publication

The paper is very well written, largely error free and presents an impressive overview of the existing literature on this topic.    If anything, the paper is probably a bit too long, the literature citations too exhaustive and it contains a degree of repetition within the results and discussion sections that could be eliminated.  These are only minor quibbles however and do not detract from the merits of this manuscript.

Response 1: Thank you very much. We have removed repetitions and unnecessary sections to make it shorter.

A few more detailed comments follow:

Point 2: In Figs 3,5,8 it would be helpful to have legends on the graphs showing the meaning of the coloured data points and lines rather than in the captions.

Response 2: Thank you. We have added legends on the graphs.

Point 3: In Fig. 3 there is no line between anoxic treatment data points for 2012 and 2014. Although the data for 2013 may be missing it would still be possible to have a line and explain in the caption.

Response 3: We have added legends to the graphs and changed the symbol for “Refrigerated bags” to a green triangle to make it easier to understand that this was a laboratory treatment. Both field treatments ended 2012, only the laboratory treatment continued until 2014.

Point 4: In Fig. 5,  C and D categories  are referred to in the caption but do not appear on the figures?

Response 4: Thank you, this was extremely clumsy, and they are now added. However, we decided to put all “fossilizable” together, as recent literature has shown that there is a continuum of sensitivity that makes it better not to use the division into “R” and “S”. This division was of little use in an experiment lasting six years anyway, it concerned geological time scales.

Point 5: There should be a legend in Fig. 5.

Response 5: We have added legends in all figures now. Thank you, this is better.

Point 6: In Fig. 7, category  C (refrigerated) referred to in the caption isn’t present as a figure?

Response 6: Thank you again, we had the figure, and don’t know what happened. It is there now.

Point 7: There are no lines on the y axes in fig. 8

Response 7: We decided to replace figure 8 with a table (Table 3).

Point 8: The use of literature citations in the results section (e.g. lines:  401-402, 457, 536, 553-554, ) is not good practise and these (along with the text associated with them) should be in the methods or discussion sections.

Response 8: Yes, thank you. We have moved them to the discussion section.

Point 9: There seems to be quite a bit of overlap between sections 3.3 and 4.8.

Response 9: We have moved sections from results to discussion to avoid repetition.

Point 10: Shouldn’t the genera and species names in the headings in section 4.8 be italicised?

Response 10: Yes, thank you, they are now.

Point 11: Describing toxic species as “evil” (lines 401-402) is taking anthropomorphism a bit far

Response 11: Yes, that has been removed.

Reviewer 2 Report

The authors of the study aim to quantify survival of dinoflagellate resting stages in two different environments, an anoxic versus oxygenated site. The main contribution of the study is its attempt to quantify the degradation rates of the entire dinoflagellate cyst community over several years in the field. Although the effect of oxygen on cyst degradation / survival is well known for many taxa there is clearly a need for more accurate quantitative data. The chosen experimental set-up of deploying plankton screen bags filled with dinoflagellate cysts at an anoxic site is a strength of the study because of its simplicity. Yet, this approach has also several disadvantages. First, the strong manipulation of the cyst community (e.g., by using a “cyst concentration apparatus”) likely affects germination and survival of different taxa and makes it difficult to compare the results of this study with the literature. Furthermore, the exclusion of grazers, which play a key role for the fate of cysts in natural environments, does not allow a thorough assessment of cyst survival rates.

Areas of weakness in this study are the lack of a clearly formulated hypothesis and assessment of environmental parameters during the deployment of (e.g., oxygen measurements at the anoxic and oxygenated site, organic content of the sediment etc. should be provided). Furthermore, standardized methodology was not used, which makes it difficult to interpret and compare the results correctly with other studies.

In general, the manuscript is not written in a clear, concise way. Some statements in the introduction are misleading and inaccurate. Recent literature (less than 5 years old) is mostly missing, many self-citations occur and new relevant literature in the field was not included. Overall, the manuscript does not seem to be scientifically sound, and the results are likely not reproducible based on the details given in the methods. I do appreciate the effort of the authors to undertake such an extensive survey, but due to too many shortcomings I did not include more specific comments in my review. 

Specific comments:

Title and the entire text should be more concise. The manuscript should have a clear focus, include hypothesis, refer to more recent similar studies and highlight the novelty of the study.

Abstract: Could be shorter, more focused and include one or two sentence about the background of the study. Why is it important to study this? What is the novelty of the results? 

Introduction: Abbreviations need to be explained the first time mentioned. Should be more concise.

Figures: Some axis labels are missing; the number of replicates should be indicated in the caption. Including a legend with the figures would improve readability. Fig.8: one data point is hidden, legend labels should be simple and clear. E.g., which treatment is considered as control?

Author Response

Point 1: The authors of the study aim to quantify survival of dinoflagellate resting stages in two different environments, an anoxic versus oxygenated site. The main contribution of the study is its attempt to quantify the degradation rates of the entire dinoflagellate cyst community over several years in the field. Although the effect of oxygen on cyst degradation / survival is well known for many taxa there is clearly a need for more accurate quantitative data. The chosen experimental set-up of deploying plankton screen bags filled with dinoflagellate cysts at an anoxic site is a strength of the study because of its simplicity. Yet, this approach has also several disadvantages. First, the strong manipulation of the cyst community (e.g., by using a “cyst concentration apparatus”) likely affects germination and survival of different taxa

Response 1: The cyst community was sampled in a cold season from black anoxic sediment, transported and stored cold. Cyst concentration was performed by settling of cysts on a slanted screen in a large basin right after sampling (we consider this a mild treatment, not a strong manipulation). The water and the room were cold and it was performed in darkness. The concentrated cyst fraction was stored cold and dark. Placement of the cold concentrated sediment containing cysts into bags was made with all lights turned off in the laboratory and bags were immediately placed dark and cold. Everything was kept on ice under black plastic as much as possible in this dim room (dim light from windows far away in late October. This is described in M&M.

Point 2: and makes it difficult to compare the results of this study with the literature there is no comparable literature.

Response 2: Nobody has published anything comparable.

Point 3: Furthermore, the exclusion of grazers, which play a key role for the fate of cysts in natural environments, does not allow a thorough assessment of cyst survival rates

Response 3: We here assess survival rates without grazers, having studied grazing on dinoflagellate resting cysts extensively earlier in very many publications. We discuss the additional effect of grazers.

Point 4: Areas of weakness in this study are the lack of a clearly formulated hypothesis and assessment of environmental parameters during the deployment of (e.g., oxygen measurements at the anoxic and oxygenated site, organic content of the sediment etc. should be provided).

Response 4: None of the reviewers had problems understanding that we wanted to compare dinoflagellate cyst preservation over time in different environments. The null hypothesis was that there would be no differences between treatments or over time. This was a multi-year study comparing cyst preservation in a well-known anoxic site with black fine-grained sediment with a site known to be oxygenated. These environments were compared to cold storage in the laboratory. Measurement of oxygen levels before the study would say little because these levels were likely to vary a lot over seasons and years, and we had no possibility to perform continuous measurements. Since the sediment was treated to carefully concentrate the cysts more than 300x, and then placed in bags with 20µm mesh size, the organic content of the original sediment, or the concentrated cyst fraction, is irrelevant. Anything smaller than 20 µm could pass freely in or out of the bags.

Point 5: Furthermore, standardized methodology was not used, which makes it difficult to interpret and compare the results correctly with other studies.

Response 5: There are no comparable studies. We wanted to study living material over time and no other such studies have been published.

Point 6: In general, the manuscript is not written in a clear, concise way. Some statements in the introduction are misleading and inaccurate.

Response 6: It is likely that we do not agree on everything, we do not intend to write misleading and inaccurate.

Point 7: Recent literature (less than 5 years old) is mostly missing

Response 7: Reviewer 3 gave us many references, and updates are now integrated.

Point 8: many self-citations occur

Response 8: We have performed and published many relevant experiments over more than twenty years. It is common and relevant with self-citations when one project leads to another with a logical connection.

Point 9: and new relevant literature in the field was not included

Response 9: we have updated this.

Point 10: Overall, the manuscript does not seem to be scientifically sound

Response 10: According to reviewer 1 and 3 it is.

Point 11: and the results are likely not reproducible based on the details given in the methods

Response 11: They are, but it would probably be difficult to find funding for it, or someone prepared to spend that much time with it, therefore the results are valuable.

Point 12: I do appreciate the effort of the authors to undertake such an extensive survey, but due to too many shortcomings I did not include more specific comments in my review.

Response 12: It is difficult to respond to this. The other two reviewers were very positive about our study. We understand and appreciate that reviewers have different interests and different points of view.

Specific comments:

Point 13: Title and the entire text should be more concise. The manuscript should have a clear focus, include hypothesis, refer to more recent similar studies and highlight the novelty of the study.

Response 13: We focus on replying to the helpful and positive comments by the two other reviewers and follow their advice. This is a unique study and we are proud of it.

Point 14: Abstract: Could be shorter, more focused and include one or two sentence about the background of the study. Why is it important to study this? What is the novelty of the results?

Response 14: For someone who is interested in the subject it is both interesting and new.

Point 15: Introduction: Abbreviations need to be explained the first time mentioned

Response 15: Thank you, they are now.

Point 16: Should be more concise.

Response16: We think it is now.

Point 17: Figures: Some axis labels are missing; the number of replicates should be indicated in the caption. Including a legend with the figures would improve readability. Fig.8: one data point is hidden, legend labels should be simple and clear.

Response 17: We have looked thoroughly into this; it seems like some figures even disappeared, which is bad and not intentional. It is corrected now.

Point 18: E.g., which treatment is considered as control?

Response 18: Originally, the samples stored in the laboratory were regarded as some kind of control, but time affected all treatments (as we knew it would). It is not easy to figure out how a true control treatment would be made or kept. Start values may be the best comparison for “values frozen in time”. We think it is informative to see the changes over time and to compare the different treatments. Both what happens in nature and during storage in a laboratory is important.

Reviewer 3 Report

Hereby I return the review of the Manuscript: Preservation of dinoflagellate cysts in different Oxygen Regimes; differences in cyst survival between toxic and anoxic natural environments.

This manuscript documents the results of a very nice elegant experiment to study the selective degradation of dinoflagellate cysts in different redox environments by placing sedimentary material with a known dinoflagellate cysts association in aerobic and aerobic depositional settings. It is the first experiment that investigates the effect of redox state on the survival of dinoflagellate cysts. Although the experimental setup as well as the data are sound and straightforward and the English spelling and grammar are good, I haven major comments on the style of the manuscript as well as the interpretation and documentation of the data.

The experiment itself is sound and the data of high quality. It is however a pity that the authors do not show all the data as there are many new aspects in this study that are worth to be shared with a large audience. It will form an important contribution for insight into depositional processes influencing dinoflagellate bloom dynamics as well as is relevant for the dinoflagellate cyst research community using cyst to establish past environmental conditions. 

However, unfortunately the manuscript is not publishable in its current form. My major concerns are focussed on the results and discussion parts that have to be completely rewritten. Therefore I did not provide detailed comments on the individual chapters but constrained myself to the most important points that have to be adapted.

I look forward to see a revised manuscript of this experiment published.

My major concerns are:

  • Both in the text and discussion the more recent literature about the subject have not been taken into account e.g. recent literature about dinoflagellate life cycles is missing (e.g. Bravo and Figueroa 2014 Microorganisms 2:1, 11-3) literature published by Mertens, Versteegh, Zonneveld etc. The data fit well with results of these more recent publications and extent the current state of the knowledge by information about hatching rates. A detailed comparison between the results of this study and that of the mentioned studies is however required.

  • The authors divide the cyst species into two main groups: “sensitive fossilizable” and “resistant fossilizable”. Apart from the fact that it is not stated in the material /method part which species are included in which group, literature published in the last decade has shown that the sensitivity of dinoflagellate cysts to post-depositional aerobic degradation is gradual ranging from species that already degrade in suboxic conditions towards extremely recalcitrant species see. e.g. Gray, et al., 2017, Review of Palaeobotany and Palynology 247, 175-187, Zonneveld et al., 2019 Marine Geology 408, 87-109. The results of this study have to be given in more detail (degradation rates of individual species and not of groups). A detailed comparison with other studies is needed. 

  • The authors state that the cyst wall of dinoflagellates is made of “dinosporin”. Organic geochemical studies have revealed however that the wall of dinoflagellate cysts has a polysaccharide backbone: see e.g. Versteegh et al., 2012 Organic Geochemistry 43, 92-102, Bogus et al., 2014: Journal of Phycology 50:2, 254-266, Mertens et al, 2017, Palynology 41: 183-202, Versteegh et al., 2020 Biogeosciences 17, 3545-356 etc. Information about the molecular characteristics of dinoflagellate cysts have to be taken into account.

  • In the discussion part the authors give a sort of “review” about the factors influencing the degradation of particulate organic matter (to which dinoflagellate cysts belong. Apart from the fact that part of this information is “outdated” and not up to date to the current insights in POC degradation processes, giving such a review is not within the scope of this study that documents results of an degradation experiment. A good overview can be found in Middelburg 2018, Biogeosciences 15:2, 413-427 and Middelburg 2019: Marine Carbon Biogeochemistry…springer. I would like to suggest that the authors list in a short paragraph which processes affect POM degradation with reference to the up to date literature and than discuss in the discussion part only those processes that are relevant for this experiment.

  • The Manuscript contains a lot of repetition especially with respect of the effect of grazing on the post-depositional degradation of dinoflagellate cysts. With respect of the experimental setup, grazing is not relevant as grazers were not able to reach the experimental material that was stored in bags with pore sizes of 20 micrometer. Repetition in general has to be reduced and the discussion about grazing should be reduced to the minimum 8see also point before)

  • The authors provide information about the degradation of a few selected species only. Since the degradation rate of dinoflagellate cysts is species specific and differs largely between species it is of importance to depict the concentration and relative abundance over time of all species which than later can be arranged along a gradient of vulnerability based on the results of this study. Including this information will enlarge significantly the value of this study.

  • The figures depicting the results are not always consistent in their axis legends and sometimes the legend is missing (e.g. fig. 7 and 8). Furthermore the figures take much space. Results of the concentrations and relative abundance data of all species can be included in one figure by placing the species below each other with at the left side depicting the concentration and at the right side the relative abundance. 

  • The new aspect in this study is the determination of the survival rates of cysts based on hatching experiments. Unfortunately not much detail is given which dinoflagellate cyst groups had high survival rates and which not. Neither does the manuscript provide details how the distinguished groups of motiles were being defined….. The number of groups over time does not provide that this information. Also figure  8a is not informative. More information is required here as this would very much strengthen this study. I would like to suggest the authors provide per dinoflagellate group a figure with absolute numbers and relative amounts of full and empty cysts over time. 

  • It is not clear what statistics have been performed and how the error bars in the figures have been established …with which program….what calculations have been used….did the authors include the error of point counting ….betainvers? … was an anova established…. (see e.g. van der Plas 1965, Howard 1998). The authors state that there was much variation of cyst concentration/association composition between the bags exposed to aerobic conditionals compared to those of the anaerobic experiment. However, the error bars given in the figures are often larger for the results of anaerobic treatment compared to those of the aerobic treatment of the same year (see e.g. year 2). This is in contrast to the conclusions of the authors. However, this 

  • The manuscript is very long (partly because of repetition and discussion of for this experiment unrelevant processes) and should be shortened.

-  The calcareous dinoflagellate cysts with and without spines are not the same species but according to the figures at least 4 different species are included in this group. Since degradation is to always found to be species specific, information about the individual species has to be provided. 

  • By preparing the material previous to the experiment (previous to homogenizing the sediments and dividing it to the experiment bags) the material was exposed to light. The material was furthermore sonificated and washed with filtered seawater under normal aerobic lab conditions. This automatically implies that the cysts in the sediments have been exposed to aerobic conditions, sonification and light previous to be placed in aerobic and anaerobic conditions. It is well known that a short exposure time of cysts to these conditions might induce hatching. This might explain the relative high hatching rates in the blancs and anaerobic environment and should be taken into accounts by interpreting the data. Furthermore the blancs have been stored at 4°C. In natural environments many bottom water environments have similar temperature regimes and it is well known that the hatching of cysts can occur also at such low temperatures. Also early diagenetic processes is not prevented by cooling the material and not freezing. This has to be taken into account by interpreting the results of the blancs and should be discussed in the discussion part. 

  • As far as I can understand from the mat/met chapter 2.1.8. germination experiments have not been carried with a standardized procedure but “slurries” have been examined for 4 to 10 days…… is it possible than to compare the outcome? If yes….how did the motile association composition (on group level) change over time.

Some selected comments on individual Chapters

Introduction

The introduction can be shortened. It contains many statements that are not completely correct. e.g.

line 48-49 …zygote transforms into a resting cysts and sinks to the seafloor to overwinter…..

dormancy period can differ from several hours to several months and many cysts hatch before reaching deeper waters especially in the subtropics/tropics (see e.g. Zonneveld 2021, 2022 https://doi.org/10.1080/09670262.2021.1885066, doi: 10.3389/fmars.2022.915755 and references therein)

54-55. geological record….comprising only of resistant resting cysts…..  Downcore presence of dinoflagellate remains is depending on the depositional environment. In anoxic environments (e.g. oxygen minimum zones) cysts that are very vulnerable to early diagenetic decay can preserve (. In exceptional depositional environments dinocysts of motile dinoflagellates can preserve (e.g. Versteegh 2004 Organic Geochemistry 35 1129–1139)

58-59. Cyst of Gymnodinium catenatum  preserve well

60-61 …..cyst possessing a thick resistant wall…. the thickness of the wall is not relevant for it’s preservation potential but it’s composition (e.g. thick walled cysts of heterotrophic dinoflagellates can be among the most vulnerable species and think walled photo/mixotrophic species among the most resistant against early diagenetic degradation)

83-84 ….. anoxic places are often holes/deep places where fine-grained sediments….. This is incorrect, there are many places in the ocean where anoxic conditions prevail (e.g. oxygen minimum zones in upwelling and high productivity regions, highly polluted (coastal) area’s, area’s where brine conditions exist).

86-87 …. aerobic bacteria are more efficient decomposers…… I think the authors refer to “microbial degradation” (by the way….bacteria and Archean are two kingdoms …. at several positions in the text it seems as if the authors imply that archaea are bacteria……)

93-94 … this is not the case, in many anaerobic regions several high concentrations of calcareous dinoflagellate cysts are b being registered (see e.g. literature by Link et al., Wendler et al)

Material/Methods……shorten to 1/3 of length, name cruise report, describe statistical methods used, calculations made, avoid repetition. Restrict to the relevant information. Avoid mentioning results or including discussions in this chapter

Results

Reduce to the essential information (e.g. Table 2…define the motile groups in mat/met….remove this to additional information, Figure 8A/B are not informative at al… 8a can be removed, 8b  )

Please provide information about the concentrations of cysts of the registered species with and with cell content over time. Group the more rare species into “other photo/micotrophic” or “other heterotrophic”.

Please provide information about the relative abundances of the individual species over time in the sediments

Please provide information about the relative abundances of the the dinoflagellate motile groups in the slurry samples over time

Is there a difference in the degradation rates of the Echinidinium species and the Protoperidinium species?

Did the authors execute hatching experiment with individual cysts that proof that Kryptoperidinium sp. is hatched from the grey-brown round cyst?

The depicted calcareous cysts “with and without crystals” belong to different species (see papers by Monteresor, Versteegh, Vink, Wendler, Willems, Meier etc.)

Please provide the results of all the species and describe the main trends.

….. discussion…. shorten and concentrate on the outcome of this experiment and the consequences it hat to studies in the research field.

Author Response

Point 1: This manuscript documents the results of a very nice elegant experiment to study the selective degradation of dinoflagellate cysts in different redox environments by placing sedimentary material with a known dinoflagellate cysts association in aerobic and aerobic depositional settings. It is the first experiment that investigates the effect of redox state on the survival of dinoflagellate cysts. Although the experimental setup as well as the data are sound and straightforward and the English spelling and grammar are good, I haven major comments on the style of the manuscript as well as the interpretation and documentation of the data.

Response 1: We are very grateful to this reviewer. The comments are kind, constructive, and important. We also received many important references. We have tried to do our best to improve the manuscript according to all comments and suggestions.

Point 2: The experiment itself is sound and the data of high quality. It is however a pity that the authors do not show all the data as there are many new aspects in this study that are worth to be shared with a large audience. It will form an important contribution for insight into depositional processes influencing dinoflagellate bloom dynamics as well as is relevant for the dinoflagellate cyst research community using cyst to establish past environmental conditions.

Response 2: It is important that the large table (Table 1) with results becomes easily available. It is however very large and we suppose it has to be supplemental. We have now added the total number of cysts per bag as the last row (including the multiplication factor from the actual count to the entire bag). Doing this, we discovered an error that had caused a huge variation in one of the three laboratory (test tube) samples for 2014. All calculations and figures are now redone with the correct values for this sample.

In Table 2 we added a column showing the percentage of the species of the cyst community in the concentrated sediment, as the average of the three replicates and with standard deviation. Many species were represented with very few individuals and the variation was quite large, therefore numbers are given as <1%, <2%, <3%, and for those over 3% of the cyst community averages are given. It does not give meaningful graphs when “zeroes” are present and with a large variation in low numbers, therefore we had to pick out the most common species and sometimes look at groups instead of species.

Point 3: However, unfortunately the manuscript is not publishable in its current form. My major concerns are focussed on the results and discussion parts that have to be completely rewritten. Therefore I did not provide detailed comments on the individual chapters but constrained myself to the most important points that have to be adapted.

Response 3: Thank you very much for all your comments! We are very grateful. We also wish to thank you for all the references. It is important to be updated and try to incorporate all that is known.

Point 4: I look forward to see a revised manuscript of this experiment published.

My major concerns are:

  • Both in the text and discussion the more recent literature about the subject have not been taken into account e.g. recent literature about dinoflagellate life cycles is missing.

Response 4: Thank you for this comment. Dinoflagellates constitute a large proportion of total protists in all size classes from pico- to microplankton and are one of the major phytoplankton groups in both marine and freshwater environments globally. They are evolutionarily very old, and the phylum occupies different niches and exemplifies vastly different life strategies. We have now tried to point this out clearly in the introduction. Only a small proportion of the dinoflagellates produce recognizable resistant resting cysts in sizes convenient to study, i.e. we cannot claim to study all dinoflagellates or say that they all have the same life strategy. We have chosen to study the “common” ones, those that can be found in the literature and are studied by other scientists. We have tried to make this clear now. Most of our past research has been focused on life cycles, the sexual life cycle of cyst producing dinoflagellates, including detailed studies on the swimming behavior of these different life stages. This interest originated from observations made during resting cyst production for grazing experiments. You are right that we have to point out that our life cycle description is a generalization, valid not to all dinoflagellates, but to the majority (or all) species in this study.

Point 5: (e.g. Bravo and Figueroa 2014 Microorganisms 2:1, 11-3.

Response 5: This paper contains a lot of interesting information, especially about the formation of the pellicular layer of pellicle cysts. We however have some concerns with the study referred to as showing asexual resting cysts; cells were assumed to be haploid, but the results could also be interpreted as showing diploid cells dividing into haploid cells that formed gametes (therefore the unusually quick growth) that mated to form sexual cysts. Behavioral observations were not made/reported (difficult at 3°C and very low light?). Motile stages of dinoflagellates have very large nuclei easily seen in LM whereas the resting cyst nucleus cannot be seen at all in LM without coloring, it is much smaller. We have worked a lot with flow cytometry and also used staining, it is not convincing to draw conclusions about ploidity by staining and comparing large and small nuclei. It does not feel comfortable for us to criticize this, but we are comfortable with using the “old-fashioned” view of the life cycle as a good generalization – with the acknowledgment of high variation within the phylum. Also, much-cited literature describes complicated life strategies in laboratory cultures where for example zygotes go back to division into vegetative cells etc. Also we have seen that in laboratory cultures many things can happen when conditions are manipulated. In nature, these things can happen too if they ever happen in the lab, of course. Recently, I saw a flower on one of my apple trees, I have never before seen this in late September. Sadly though, I don’t think this flower will give me an apple, not even if my bees are happy to find it. Nature is harsher than the laboratory sometimes (or often). Seasons tend to come in consecutive order. Zygotes that form resistant resting stages that can overwinter have a brighter future than those who divide again into vegetative cells right before conditions become impossible to survive for motile cells. In nature, that is, in the lab the latter might become those dominating the culture… and the interpretation of experiments made with it)

Point 6: literature published by Mertens, Versteegh, Zonneveld etc. The data fit well with results of these more recent publications and extent the current state of the knowledge by information about hatching rates. A detailed comparison between the results of this study and that of the mentioned studies is however required.

Response 6: Thank you for all the references! They are important and we have included them where they are relevant.

Point 7: The authors divide the cyst species into two main groups: “sensitive fossilizable” and “resistant fossilizable”. Apart from the fact that it is not stated in the material /method part which species are included in which group, literature published in the last decade has shown that the sensitivity of dinoflagellate cysts to post-depositional aerobic degradation is gradual ranging from species that already degrade in suboxic conditions towards extremely recalcitrant species see. e.g. Gray, et al., 2017, Review of Palaeobotany and Palynology 247, 175-187, Zonneveld et al., 2019 Marine Geology 408, 87-109. The results of this study have to be given in more detail (degradation rates of individual species and not of groups). A detailed comparison with other studies is needed. 

Response 7: Thank you very much! We have removed the division of fossilizable cysts into groups, this was not relevant to a short study as ours anyway. Since some of the species were very few, we had to group them in order to understand, analyze and describe the results. All results and analyses were revisited, and we think that the choices of graphs are those that show the general picture most clearly. You are right that more results deserve attention, but these will have to be published separately to not divert from the general results.

Point 8: The authors state that the cyst wall of dinoflagellates is made of “dinosporin”. Organic geochemical studies have revealed however that the wall of dinoflagellate cysts has a polysaccharide backbone: see e.g. Versteegh et al., 2012 Organic Geochemistry 43, 92-102, Bogus et al., 2014: Journal of Phycology 50:2, 254-266, Mertens et al, 2017, Palynology 41: 183-202, Versteegh et al., 2020 Biogeosciences 17, 3545-356 etc. Information about the molecular characteristics of dinoflagellate cysts have to be taken into account.

Response 8: Thank you. We are grateful for the comments and important additional reading. We have removed the words “sporopollenin” and “dinosporin”, and added two references regarding the cyst wall.

Point 9: In the discussion part the authors give a sort of “review” about the factors influencing the degradation of particulate organic matter (to which dinoflagellate cysts belong. Apart from the fact that part of this information is “outdated” and not up to date to the current insights in POC degradation processes, giving such a review is not within the scope of this study that documents results of an degradation experiment. A good overview can be found in Middelburg 2018, Biogeosciences 15:2, 413-427 and Middelburg 2019: Marine Carbon Biogeochemistry…springer. I would like to suggest that the authors list in a short paragraph which processes affect POM degradation with reference to the up to date literature and than discuss in the discussion part only those processes that are relevant for this experiment.

Response 9: Thank you very much! We have removed a lot of text and shortened these sections. It is still important that enough information remains to make clear that the sediment itself disintegrates since biologists seldom realize this. As Middelburg points out, scientists from different areas each make mistakes, forgetting important things that they know nothing about (since it isn’t their field, but self-evident for someone in a different field of research).

Point 10: The Manuscript contains a lot of repetition especially with respect of the effect of grazing on the post-depositional degradation of dinoflagellate cysts. With respect of the experimental setup, grazing is not relevant as grazers were not able to reach the experimental material that was stored in bags with pore sizes of 20 micrometer. Repetition in general has to be reduced and the discussion about grazing should be reduced to the minimum 8see also point before)

Response 10: Thank you. We have tried to remove repetition and reduced the discussion about grazing.  It is however relevant to discuss grazing for understanding the natural environment of dinoflagellate cysts.

Point 11: The authors provide information about the degradation of a few selected species only. Since the degradation rate of dinoflagellate cysts is species specific and differs largely between species it is of importance to depict the concentration and relative abundance over time of all species which than later can be arranged along a gradient of vulnerability based on the results of this study. Including this information will enlarge significantly the value of this study.

Response 11: We have looked into this, but many species have low abundance. Since we have three replicates for every treatment each year, we also have information about the variation – which tells us how common a species must be within the cyst assembly for us to be able to say anything with significance. We see that with time the unfossilizable species become uncommon – the living unknowns disappear with time and beautifully clean and empty well-known microfossils appear, those that can be found in the literature. Even though we would have liked to be able to, for two reasons we cannot provide a series of species arranged in order of vulnerability: 1) they are too few and numbers vary too much when they are rare 2) the experimental time was six years only, which is nothing for a microfossil as it seems – unless it is exposed to oxygen. With oxygen, everything disappeared within short time and no order or gradient of disappearance could be established.

Point 12: The figures depicting the results are not always consistent in their axis legends and sometimes the legend is missing (e.g. fig. 7 and 8). Furthermore the figures take much space. Results of the concentrations and relative abundance data of all species can be included in one figure by placing the species below each other with at the left side depicting the concentration and at the right side the relative abundance.

Response 12: We have added legends and removed unnecessary figures. We made figures in very many different ways, using different statistical programs, during the analysis of the data. Sometimes an idea is good, but the figure becomes messy and difficult to understand. Some species-specific results will have to be lifted out and published separately.

Point 13: The new aspect in this study is the determination of the survival rates of cysts based on hatching experiments. Unfortunately not much detail is given which dinoflagellate cyst groups had high survival rates and which not.

Response 13: Germination experiments were made to check whether live-appearing cysts in permanent cyst mounts could be regarded as alive or not.

Point 14: Neither does the manuscript provide details how the distinguished groups of motiles were being defined…..

Response 14: In Table 2 the appearance as vegetative cell in slurry culture is given (at 400x magnification), as identified by standard available literature for vegetative cells of dinoflagellates.

Point 15: The number of groups over time does not provide that this information. Also figure  8a is not informative.

Response 15: We removed figure 8a and inserted a short table instead of figure 8b (Table 3 now).

Point 16: More information is required here as this would very much strengthen this study.

Response 16: We can only provide information about germinated dinoflagellates observed in slurry cultures at 400X magnification, and a few more detailed observations year 2014. This is discussed below with more detailed comments.

Point 17: I would like to suggest the authors provide per dinoflagellate group a figure with absolute numbers and relative amounts of full and empty cysts over time.

Response 17: This is a good suggestion, and you convince us that we have to write additional papers based on the extensive results we have. All aspects cannot be covered here, we need to show the general picture and changes that are statistically significant. 

Point 18: It is not clear what statistics have been performed and how the error bars in the figures have been established …with which program….what calculations have been used….did the authors include the error of point counting ….betainvers? … was an anova established…. (see e.g. van der Plas 1965, Howard 1998).

Response 18: Thank you. Of course, of course, this is how we know we have results. We have used Excel, StatGraphics and R with ggPlot2. Error bars are standard error of the mean. The significance level was always 0.05. Many Anovas have been made.

Point 19: The authors state that there was much variation of cyst concentration/association composition between the bags exposed to aerobic conditionals compared to those of the anaerobic experiment. However, the error bars given in the figures are often larger for the results of anaerobic treatment compared to those of the aerobic treatment of the same year (see e.g. year 2). This is in contrast to the conclusions of the authors

Response 19: This is a misunderstanding. We try to say that the anoxic macro-environment in the field was homogenous, that there were no differences between poles, but the variation found was intrinsic within the large original sample before it was divided into all the bags. This is an example of when Anovas were useful – it showed us that the variation was not between poles within the environment, but between bags within the environment = within the original sample before it was divided into bags (because of cyst clumping at a microscopic level, since the concentrated sediment was carefully treated before placement in bags). The oxygenized macro-environment was variable. We have now added “macro-environment”. We do not say that variation in the results were larger in the oxygenized environment, we say that there was a variation between the bags, within each environment since cysts are sticky etc and appear in clumps. This variation is something that must be present in all cyst investigations, but most studies are un-replicated and variation thus not seen.

However, this   

text is missing here.

Point 20: The manuscript is very long (partly because of repetition and discussion of for this experiment unrelevant processes) and should be shortened.

Response 20: Thank you. Unnecessary parts and repetitions have now been removed.

Point 21:  The calcareous dinoflagellate cysts with and without spines are not the same species but according to the figures at least 4 different species are included in this group. Since degradation is to always found to be species specific, information about the individual species has to be provided.

Response 21: We had to consider Scrippsiella spp. as a group of species here. Our past experience from many experiments on cyst formation using clonal cultures of Scrippsiella lachrymosa has shown us that one clone can give rise to resting cysts with widely different crystal structures, as well as cysts without crystals. In addition, in the many grazing studies, we have seen calcarous cysts losing crystals after passage through animal guts. We have also repeatedly observed cysts from acidic environments without crystals. The literature is in accordance with this. Dissolution of crystals must happen over time (and, as you say, be species-specific), and we cannot know how long time the cysts had spent in the sediment before we took them and used them in the experiment, certainly they were of different ages. We have changed the text to reflect this view, we cannot know if cysts with and without crystals were the same or different species in this experiment. We cannot know from this experiment if the crystals have anything to do with survival, if they offer any protection at all, and we do not discuss it.

Point 22: By preparing the material previous to the experiment (previous to homogenizing the sediments and dividing it to the experiment bags) the material was exposed to light.

Response 22: No, it was not. The concentration was performed carefully in cold and darkness.

Point 23: The material was furthermore sonificated and washed with filtered seawater under normal aerobic lab conditions.

Response 23: No, no. We have tried to be more clear in M&M. This was not made prior to placement in bags to be deployed. Sonication and washing were performed after sampling, when permanent mounts were going to be made, and slurry cultures.

Point 24: This automatically implies that the cysts in the sediments have been exposed to aerobic conditions, sonification and light previous to be placed in aerobic and anaerobic conditions.

Response 24: No, no, no, we have tried to make our text very clear on this. Thank you for misunderstanding, and in that way pointing out the possibility to misunderstand the text here. The sediment was concentrated cold and in darkness using a large basin where cysts fell on a slanted screen according to their sinking properties, and no other treatment was made. It was placed in bags as dark as possible, and cold, to prevent exposure to anything that could “wake up” cysts prior to placement in experimental conditions.

Point 25: It is well known that a short exposure time of cysts to these conditions might induce hatching

Response 25: Yes, indeed, that is why we avoided this. We hope it is more clear now.

Point 26: This might explain the relative high hatching rates in the blancs and anaerobic environment and should be taken into accounts by interpreting the data.

Response 26: After the yearly sampling, before culturing, cysts were sonicated and sieved. This was made in order to check whether live-looking cysts were alive and could germinate into vegetative cells. Cysts might have contents but be dead, we had to check this. Then, the purpose was to wake them up, and sonication and exposure to light in the lab was not considered a disadvantage.

Point 27: Furthermore the blancs have been stored at 4°C. In natural environments many bottom water environments have similar temperature regimes and it is well known that the hatching of cysts can occur also at such low temperatures

Response 27: Yes, if oxygen is available.

Point 28: Also early diagenetic processes is not prevented by cooling the material and not freezing

Response 28: Yes, things happen also during laboratory storage. We describe what happens.

Point 29: This has to be taken into account by interpreting the results of the blancs and should be discussed in the discussion part.

Response 29: We discuss this, it is important.

Point 30: As far as I can understand from the mat/met chapter 2.1.8. germination experiments have not been carried with a standardized procedure but “slurries” have been examined for 4 to 10 days…… is it possible than to compare the outcome?

Response 30: The time-consuming procedure of individually picking and germinating cysts was not possible for us with the time frame or personnel we had. The purpose of slurry culturing was to confirm that/check if/ live-looking cysts were indeed alive or not. Would we be able to trust that live-appearing cysts in the permanent mounts were really alive, or were the contents dead?

Point 31: If yes….how did the motile association composition (on group level) change over time.

Response 31: Slurry culturing is non-quantitative, only qualitative observations can be made. There are no species numbers for the vegetative cells and the only comparisons that can be made are the qualitative observations (something was seen, then it was there). Slurry culturing has advantages and disadvantages; it is an easy and nice way to check if dinoflagellate cysts from sediment samples are alive without causing them too much disturbance, but some cells germinate fast and then die since they don’t like the conditions, some germinate and divide and grow, some get eaten and some eat others. One cell does not equal one germinated cyst, and the species composition differs every day. You can never say that if something was not observed, it didn’t germinate. If you saw something, then you saw it, that is what you have and that is the advantage.

Some selected comments on individual Chapters

Introduction

Point 32: The introduction can be shortened. It contains many statements that are not completely correct. e.g. line 48-49 …zygote transforms into a resting cysts and sinks to the seafloor to overwinter…..dormancy period can differ from several hours to several months and many cysts hatch before reaching deeper waters especially in the subtropics/tropics (see e.g. Zonneveld 2021, 2022 https://doi.org/10.1080/09670262.2021.1885066, doi: 10.3389/fmars.2022.915755 and references therein)

Response 32: Thank you. These papers are very interesting, but this manuscript is concerned with a temperate region near the coast and in shallow eutrophicated waters. As you point out we should not generalize to all dinoflagellates everywhere, but make clear that we describe the general pattern for cyst-producing dinoflagellates in temperate coastal environments. We tried to make this clear now. The sediment we used was collected from black anoxic mud at the end of October. Their dormancy periods are of course species-specific, and also for most of them unknown.

Point 33: 54-55. geological record….comprising only of resistant resting cysts…..  Downcore presence of dinoflagellate remains is depending on the depositional environment. In anoxic environments (e.g. oxygen minimum zones) cysts that are very vulnerable to early diagenetic decay can preserve (. In exceptional depositional environments dinocysts of motile dinoflagellates can preserve (e.g. Versteegh 2004 Organic Geochemistry 35 1129–1139)

Response 33: This was fascinating reading! (It reminds me of a trip to the museum in Varberg where they have "the Bocksten Man"; the remains of a medieval man's body found in a bog. On the wall, they had a small showcase and the text “probably part of the upper lip”. Anyway, most of us disappear with time, dinoflagellates and people).

Point 34: 58-59. Cyst of Gymnodinium catenatum  preserve well

Response 34: Yes, and many more. We used examples common in the area and now removed "globally occurring" even though they are.  It is easier to not start discussing species that are important in places far away and are not present in this study.

Point 35: 60-61 …..cyst possessing a thick resistant wall…. the thickness of the wall is not relevant for it’s preservation potential but it’s composition (e.g. thick walled cysts of heterotrophic dinoflagellates can be among the most vulnerable species and think walled photo/mixotrophic species among the most resistant against early diagenetic degradation)

Response 35: Thank you. We removed “thick”.

Point 36: 83-84 ….. anoxic places are often holes/deep places where fine-grained sediments….. This is incorrect, there are many places in the ocean where anoxic conditions prevail (e.g. oxygen minimum zones in upwelling and high productivity regions, highly polluted (coastal) area’s, area’s where brine conditions exist).

Response 36: Thank you, we have changed the language here.

Point 37: 86-87 …. aerobic bacteria are more efficient decomposers…… I think the authors refer to “microbial degradation” (by the way….bacteria and Archean are two kingdoms …. at several positions in the text it seems as if the authors imply that archaea are bacteria……)

Response 37: Thank you. We have corrected this.

Point 38: 93-94 … this is not the case, in many anaerobic regions several high concentrations of calcareous dinoflagellate cysts are b being registered (see e.g. literature by Link et al., Wendler et al)

Response 38: Thank you. We now make clear that we mean organic-rich black mud where acids are formed that dissolve calcium carbonate.

Point 39: Material/Methods……shorten to 1/3 of length, name cruise report, describe statistical methods used, calculations made, avoid repetition. Restrict to the relevant information. Avoid mentioning results or including discussions in this chapter

Response 39: We have tried to shorten the Materials and Methods to contain only relevant information, but it is still long since no prior studies have used the same methods; it is new and it is important that it can be understood, especially since it concerns living material. Some misunderstandings by reviewers have shown that it is important to be both thorough and clear. The sole purpose of the “cruise” was this sampling, which means the report from the cruise is this report. The port was not very far from the sampling area (neighboring the NOAA laboratory), thus samples could be taken and transported to the laboratory without causing unnecessary disturbance to the living material.

Statistical methods and calculations are now mentioned.

Results

Point 40: Reduce to the essential information (e.g. Table 2…define the motile groups in mat/met….remove this to additional information, Figure 8A/B are not informative at al… 8a can be removed, 8b  )

Response 40: We removed Figure 8a and replaced 8b with a small table (Table 3 now). We removed what was Table 3 before, and now mention germinated species with unknown cysts only in the text instead.

Point 41: Please provide information about the concentrations of cysts of the registered species with and with cell content over time. Group the more rare species into “other photo/micotrophic” or “other heterotrophic”.

Response 41: Thank you for this suggestion. The material is very extensive. You help us realize that many interesting aspects deserve attention and they will have to be published separately, otherwise this paper becomes much too long. We here must focus on the main results to make the general picture as clear as possible.

Point 42: Please provide information about the relative abundances of the individual species over time in the sediments.

Response 42. During the analysis of the data, we have made graphs in many different ways, for example, this for living cysts only (see file):

It is hard to make figures that are easy to understand when there are so many samples and species, and wanting to show the variation complicates it. We will have to write additional papers.

Point 43: Please provide information about the relative abundances of the the dinoflagellate motile groups in the slurry samples over time.

Response 43: This information would definitely be wonderful to have, but sadly it is impossible with the currently available techniques. Slurry cultures are not quantitative, one can only make qualitative observations (identify what can be identified). The cells swim around and cannot be counted, they divide, die, and eat each other. One cell does not equal one germinated cyst. Some species germinate quickly, some slowly, some divide, and some die. It is living chaos and the cells are caught in an environment that does not fit them for long. We made other experiments before, with different sediment and germinated it in different environments (using slurry cultures) and followed germination. We used several methods, for example, FlowCam, and saw many interesting things. But the experiment lasted only two weeks, and at least a year would have been needed for methods development and building of picture libraries, etc.

Point 44: Is there a difference in the degradation rates of the Echinidinium species and the Protoperidinium species?

Response 44: Since there were few individuals of each species and variation between replicates, we could not see this. In Figure 6, there is a slight decrease for Protoperidinium, but not for the Diplopsalis group (with Echinidinium spp.). However, you see that the error bars are overlapping and the differences are not significant. Six years is probably nothing for these cyst walls unless animals eat and crush them into small fragments that we cannot find or identify. We will look further into living versus dead over time for each species and group as you have suggested, but this takes time that we do not have now and would take up too much space in this manuscript.

Point 45: Did the authors execute hatching experiment with individual cysts that proof that Kryptoperidinium sp. is hatched from the grey-brown round cyst?

Response 45: No, we are sorry about this, we wish we had. There was no time for this. AP came on travel grants for a short time every year and had very much to do, there was no time to do this or follow it up, sadly. It would have been nice to have.

Point 46: The depicted calcareous cysts “with and without crystals” belong to different species (see papers by Monteresor, Versteegh, Vink, Wendler, Willems, Meier etc.).

Response 46: Yes, we have regarded it as a group here. We have now changed the wording. We group them as Scrippsiella spp. and make clear that several different species constitute this group. They probably experience different sensitivity to the dissolution of crystals. As mentioned above, we have long experience with cyst production in the laboratory using Scrippsiella lachrymosa, and have seen it make very different crystals in the same culture. This experience, as well as observations of and reading literature on the dissolution of calcareous crystals at low pH, made AP feel very humble about species identification from crystal structure in Scrippsiella spp.

Point 47: Please provide the results of all the species and describe the main trends.

Response 47: We added the relative amount of each species in the original sediment in Table 2. Only the major species and groups could be followed with statistical significance, and this is what we have shown in the figures.

Point 48: ….. discussion…. shorten and concentrate on the outcome of this experiment and the consequences it hat to studies in the research field.

Response 48: We have now removed large parts about grazing that were not relevant, as well as repetitions. We try to focus on the outcome of the experiment and the consequences it has ecologically.

Again, we wish to thank you very much for your interest, your kind helpfulness, and all the constructive comments.

Submission Date

05 August 2022

Date of this review

19 Sep 2022 13:54:45